# Epigenetic gene expression links heart failure to memory impairment

Md Rezaul Islam[1], Dawid Lbik[2,†], M Sadman Sakib[1,†,§], Raoul Maximilian Hofmann[2], Tea Berulava[1], Martí Jiménez Mausbach[1], Julia Cha[1], Maria Goldberg[1], Elerdashvili Vakhtang[1], Christian Schiffmann[1], Anke Zieseniss[3,4], Dörthe Magdalena Katschinski[3,4] ⬤, Farahnaz Sananbenesi[5], Karl Toischer[2,3,*,‡] ⬤ & Andre Fischer[1,6,7,**,‡] ⬤

## Abstract

In current clinical practice, care of diseased patients is often restricted to separated disciplines. However, such an organ-centered approach is not always suitable. For example, cognitive dysfunction is a severe burden in heart failure patients. Moreover, these patients have an increased risk for age-associated dementias. The underlying molecular mechanisms are presently unknown, and thus, corresponding therapeutic strategies to improve cognition in heart failure patients are missing. Using mice as model organisms, we show that heart failure leads to specific changes in hippocampal gene expression, a brain region intimately linked to cognition. These changes reflect increased cellular stress pathways which eventually lead to loss of neuronal euchromatin and reduced expression of a hippocampal gene cluster essential for cognition. Consequently, mice suffering from heart failure exhibit impaired memory function. These pathological changes are ameliorated via the administration of a drug that promotes neuronal euchromatin formation. Our study provides first insight to the molecular processes by which heart failure contributes to neuronal dysfunction and point to novel therapeutic avenues to treat cognitive defects in heart failure patients.

**Keywords** cognition; epigenetics; heart failure; histone; memory impairment
**Subject Categories** Cardiovascular System; Chromatin, Transcription & Genomics; Neuroscience
See also: **G Condorelli & M Matteoli** (March 2021)

## Introduction

Traditionally, clinical medicine is organized by organ-centered disciplines which is reflected in the currently applied diagnostics and treatments of patients. This approach has been also commonly adopted in research strategies but it is becoming evident that novel interdisciplinary efforts are needed to improve therapies of complex diseases. For example, heart failure (HF) is a complex, debilitating condition afflicting millions of people worldwide (Savarese & Lund, 2017). However, in addition to the detrimental phenotypes linked directly to cardiac dysfunction, cognitive deficits present a major burden to patients with HF (Pressler et al, 2010; Hajduk et al, 2013; Ampadu & Morley, 2015; Doehner et al, 2017). Moreover, epidemiological studies have clearly demonstrated that HF significantly increases the risk for dementia and age-associated neurodegenerative diseases such as Alzheimer's disease (AD) (Angermann et al, 2012; Cermakova et al, 2015; Satizabal et al, 2016). In line with these observations, a consistent finding in HF patients is a substantially reduced cerebral blood flow (Roy et al, 2017) and imaging studies reveal subsequent structural and functional cerebral alterations including changes in key regions linked to memory formation, such as the hippocampus (Kumar et al, 2011; Pan et al, 2013; Kumar et al, 2015; Woo et al, 2015). However, how HF affects hippocampal function at the molecular level remains to be explored and thus effective therapies to manage cognitive impairment if HF patients do not exist yet. On the contrary, the therapeutic approaches currently used to treat cardiac phenotypes in HF patients lack evidence for improving cognition (Cleland et al, 2005; Frigerio & Roubina, 2005; Arnold et al, 2006) or have even been linked to an increased incidence of AD (Khachaturian et al, 2006;

1 Department for Systems Medicine and Epigenetics, German Center for Neurodegenerative Diseases (DZNE), Göttingen, Germany
2 Clinic of Cardiology and Pneumology, Georg-August-University, Göttingen, Germany
3 German Center for Cardiovascular Research (DZHK), Göttingen, Germany
4 Institute for Cardiovascular Physiology, University Medical Center, Georg-August University Göttingen, Göttingen, Germany
5 Genome Dynamics, German Center for Neurodegenerative Diseases (DZNE), Göttingen, Germany
6 Department of Psychiatry and Psychotherapy, University Medical Center Göttingen, Göttingen, Germany
7 Cluster of Excellence "Multiscale Bioimaging: from Molecular Machines to Networks of Excitable Cells" (MBExC), University of Göttingen, Göttingen, Germany
*Corresponding author. Tel: +49 551 3966318; E-mail: ktoischer@med.uni-goettingen.de
**Corresponding author. Tel: +49 551 3961211; E-mail: andre.fischer@dzne.de
† These authors contributed equally to this work
‡ These authors contributed equally to this work
§ Correction added on 05 March 2021, after first online publication: the author's name was changed from Md Sadman Sakib to M Sadman Sakib.

Pressler *et al*, 2010; Galli & Lombardi, 2014; Solomon *et al*, 2017), suggesting that HF may lead to long-lasting adaptive changes in neurons that can persist despite improvement of cardiac function. Thus, a better understanding of HF-mediated molecular alterations in neurons is of utmost importance but corresponding data are lacking. Consequently, international organizations such as the European Society of Cardiology (ESC) have recommended that cardiology and dementia research experts should team-up to identify therapeutic interventional options for managing cognitive impairment in subjects with HF (Ponikowski *et al*, 2018). In this study, we took on this challenge and show that HF leads to specific changes in hippocampal gene expression that are linked to memory impairment. Targeting aberrant gene expression via epigenetic drugs ameliorates these phenotypes suggesting a key role of this process in HF-mediated cognitive dysfunction. Moreover, our data suggest that therapeutic strategies directed toward epigenetic gene expression provide a therapeutic avenue to improve cognition in HF patients and ameliorate their risk to develop AD.

## Results

### Heart failure in CamkIIδc TG mice leads to hippocampal gene expression changes indicative of dementia

With the aim to elucidate the molecular processes by which cardio-vascular dysfunction leads to memory impairment and increases the risk for dementia, we decided to employ a well-established mouse model for HF in which cardiomyocyte-specific kinase CamkIIδc is overexpressed under the control of the alpha-MHC promoter (CamkIIδc TG mice) (Maier *et al*, 2003). Thus, overexpression of CamkIIδc is specific to cardiomyocytes and is not detected in other organs, including the brain (Maier *et al*, 2003), making it a *bona fide* model to study the impact of HF on brain function. In line with this, we confirmed that expression of the CamkIIδc transgene was absent in brain (Fig 1A). We reasoned that this well-defined genetic HF model would be superior to other experimental approaches linked for example to cerebral hypoperfusion such as carotid artery occlusion, since it allowed us to study brain function in response to the very precise and exclusive manipulation of cardiac tissue. Although CamkIIδc transgenic mice display early hypertrophy at 8 weeks of age, substantial functional and structural changes are observed at 3 months of age (Sossalla *et al*, 2011). Therefore, we investigated 3-month-old CamkIIδc TG mice, and in line with previous findings, these mice displayed HF with left ventricular dilatation, impaired ejection fraction, cardiac output, and cardiac index, while heart/body weight ratio and left ventricle/body weight ration as well as the lung/body weight ratio was increased (Fig 1B and C), whereas the overall body weight was not affected ($P = 0.863$ for CamkIIδc TG vs control mice, $n = 8$, unpaired *t*-test). As a first approach to study the impact of cardiac dysfunction on brain plasticity, we decided to analyze the transcriptome of the hippocampal CA1 region in 3-month-old CamkIIδc TG mice (Fig 1D). This was based on data showing that (i) gene expression is a sensitive molecular correlate of memory function and is deregulated in dementia patients and corresponding mouse models (Fischer, 2014a); (ii) the hippocampal CA1 regions is essential for spatial reference memory in rodents and humans and is affected early in AD (Fischer, 2014a); and (iii)

imaging data show functional changes of the hippocampal CA1 region in patients with HF (Woo *et al*, 2015). RNA-seq analysis revealed substantial changes in the CA1 transcriptome of 3-month-old CamkIIδc TG and control mice that were obvious in a principal component analysis (PCA; Fig 1D). Namely, 1,780 genes were up-regulated and 2,014 genes were down-regulated in CamkIIδc TG when compared to the control group (Fig 1E; Dataset EV1). Comparison of the differentially expressed genes to previously reported cell type-specific gene expression datasets (Merienne *et al*, 2019) revealed that up-regulated genes were linked to neurons, microglia, and astrocytes, while down-regulated genes were mainly associated with neurons (Fig 1F). Further pathway analysis showed that up-regulated genes are related to cellular stress response pathways such as oxidative and endoplasmic reticulum (ER) stress (Fig 1G, Dataset EV1), while down-regulated genes are linked to cognition, protein folding, and processes related to protein methylation (Fig 1G, Dataset EV1). We decided to confirm the RNA-sequencing data by testing differential expression for selected genes representing changes related to increased cellular stress processes, in this case "ER stress" and down-regulated processes such as "protein methylation". qPCR analysis confirmed increased expression of the ER stress-related genes Fez1, Fez2, and Bcap31 (Fig 1H). We also tested the expression of several histone 3 lysine 4 (H3K4)-specific lysine methyltransferases (Kmts), since these pathways were detected in the RNA-seq data and several of the Kmt's, such as Kmt2a, were found to be essential for memory formation (Gupta *et al*, 2010; Kerimoglu *et al*, 2013; Jakovcevski *et al*, 2015; Kerimoglu *et al*, 2017). Indeed, we observed that Kmt2a and Kmt2d were significantly down-regulated in CamkIIδc TG mice (Fig 1H). Specificity of this observation was demonstrated by the fact that other H3K4 methyltransferases such as Kmt2b and Kmt2c were not differentially expressed.

The observation that genes implicated with oxidative and ER stress are increased in the hippocampus of CamkIIδc TG mice is in line with previous findings linking HF to hypoxia as a consequence of cerebral hypoperfusion (Bikkina *et al*, 1994; Verdecchia *et al*, 2001; Perlman, 2007), although additional processes than hypoperfusion likely play a role. The concomitant down-regulation of genes linked to cognition let us to hypothesize about a potential link between the observed cellular stress-related gene expression changes and the decreased expression of genes associated with cognition. Namely, we wondered whether the decreased expression of genes linked to cognition could be a consequence of the activation of cellular stress pathways. We decided to test this hypothesis further with a focus on hypoxia and ER stress as key cellular stress pathways. Since data on the effects of hypoxia and ER stress on hippocampal gene expression at the genome-wide level are still rare, we decided to performed RNA sequencing from mixed hippocampal neuronal cultures that were subjected to either hypoxia or ER stress. First, we analyzed hypoxia. Differential expression analysis revealed a substantial amount of genes that were differentially expressed in response to hypoxic conditions (Dataset EV2). We then compared the genes up- and down-regulated in hippocampal cultures in response to hypoxia to the genes up- and down-regulated in the hippocampus of CamkIIδc TG mice. This analysis revealed a significant overlap of not only up- but also down-regulated genes suggesting that hypoxic conditions are sufficient to induce gene expression changes similar to that detected in the hippocampus of mice suffering from HF (Fig 1I). We employed the same experimental

settings to test the impact of ER stress that can be modeled via the administration of tunicamycin. Thus, RNA sequencing was performed from mixed hippocampal neuronal cultures upon treatment with tunicamycin (Dataset EV3). Our data show that genes deregulated in response to tunicamycin also significantly overlap with genes affected in CamkIIδc TG mice, although to a lesser extend when compared to hypoxia (Fig 1I). In sum, these data suggest a scenario in which HF that is linked to cerebral hypoperfusion leads to hypoxia, oxidative, and ER stress-related hippocampal gene expression changes which are upstream of the reduced expression of neuronal genes important for cognition. Taken into account that impaired expression of genes essential for cognitive function is also a key hallmark of dementia, these data provide a plausible hypothesis to explain—at least in part—cognitive dysfunction in response to HF. To provide further evidence for this hypothesis, we first retrieved published datasets in which brain-specific gene expression changes were reported in mouse models with impaired memory function, namely models for aging-associated memory decline (Benito *et al*, 2015), models for AD (Gjoneska *et al*, 2015), and fronto-temporal dementia (FTLD) (Swarup *et al*, 2018). We compared these datasets to the transcriptional alterations observed in the hippocampus of CamkIIδc TG mice (Fig 1J). Interestingly, there was a significant overlap of genes up-regulated in the hippocampus of CamkIIδc TG mice and genes up-regulated in the hippocampus of cognitively impaired old mice, in CK-p25 mice representing a model for AD-like neurodegeneration and in the cortex of FVB mice, representing a mouse model for fronto-temporal dementia FTLD (Fig 1J). Similarly, genes down-regulated in the hippocampus of CamkIIδc TG mice significantly overlapped with the genes down-regulated in models for aging, AD-like neurodegeneration and FTLD (Fig 1J).

Thus, the hippocampal gene expression signature observed in response to HF partly overlaps to the gene expression changes detected in cognitive diseases. On this basis, we hypothesized that aberrant hippocampal gene expression and especially the decreased expression of learning and memory genes could be a central process in HF mediated cognitive impairment and might therefore represent a suitable target for therapeutic intervention. To further substantialize and test this hypothesis, we decided to analyze memory function in CamkIIδc TG mice directly.

### Heart failure in CamkIIδc TG is associated with impaired hippocampus-dependent memory consolidation

Three-month-old CamkIIδc TG ($n = 16$) and control mice ($n = 13$) were subjected to behavioral testing. Importantly, when subjected to the open field test, CamkIIδc TG and control mice traveled similar distances with the same speed, indicating that explorative behavior and basal motor function is normal (Fig 2A). Both groups also spent similar time in the center of the open field arena, suggesting that anxiety behavior is not affected in CamkIIδc TG mice (Fig 2A). Subsequently, mice were subjected to the Barnes Maze, a hippocampus-dependent spatial navigation-learning test (see Materials and Methods for details). Two-way ANOVA analysis revealed that CamkIIδc TG mice spent significantly more time to find the escape hole when compared to littermate controls (Fig 2B). These data suggest that hippocampus-dependent memory function is impaired in CamkIIδc TG mice. A detailed analysis of the different strategies

to find the escape hole confirmed this observation and revealed that in comparison with control mice, CamkIIδc TG mice failed to adapt hippocampus-dependent strategies (direct, short, and long chaining approaches), which are generally considered to depend on higher cognitive abilities than the other strategies (Fig. 2C). To quantify this observation, we calculated the cumulative strategy score (see Materials and Methods for details) that was significantly reduced in CamkIIδc TG mice when compared to the control group (Fig 2D), further confirming that CamkIIδc TG mice exhibit impaired hippocampus-dependent learning abilities. We also assayed memory retrieval 24 h after the final day of training by placing the mice into the Barnes Maze arena with the escape hole being closed and measured the visits to the escape hole. The number of visits to the escape hole during the 120-s test period was significantly lower in CamkIIδc TG mice when compared to the control group, indicating impaired retrieval of spatial memories (Fig 2E). In summary, these findings are in line with our gene expression data (See Fig 1) and show that CamkIIδc overexpression-induced HF leads to cognitive deficits. To substantiate these observations, we analyzed an additional non-genetic model of HF, namely the myocardial infarction (MI) model. Our data show that MI mice also develop hippocampus-dependent memory impairments and exhibit a severe deregulation of hippocampal gene expression that is similar to our data obtained in CamkIIδc TG mice (Fig EV1; Dataset EV4).

### Heart failure-related down-regulation of hippocampal genes is linked to reduced neuronal H3K4 methylation

The finding that CamkIIδc mice indeed exhibit memory impairments allowed us to move on and explore our hypothesis that decreased expression of hippocampal learning and memory genes might be one of the underlying mechanisms by which HF leads to cognitive decline. Our gene expression data suggest that genes down-regulated in the hippocampal CA1 region of CamkIIδc TG mice mainly reflect neuron-specific changes (See Fig 1F). In addition to pathways related to "cognition", a major molecular process linked to these genes was protein methylation including down-regulation of H3K4-methlytransferases such as Kmt2a (see Fig 1H). Since reduced neuronal expression of Kmt2a and corresponding genome-wide reduction of H3K4me3 has been linked to memory impairment and AD (Gjoneska *et al*, 2015; Kerimoglu *et al*, 2017), these data point to the possibility that altered H3K4-methylation may—at least in part—underlie the observed down-regulation of neuronal genes in CamkIIδc TG mice. To test this hypothesis, we retrieved and re-analyzed hippocampal RNA-seq data from mutant mice that lack the H3K4 methyltransferases Kmt2a or Kmt2b from hippocampal neurons of the adult brain and also display hippocampus-dependent memory impairment (Kerimoglu *et al*, 2013; Kerimoglu *et al*, 2017). Our data reveal that genes decreased in CamkIIδc TG mice show a significant overlap with the genes affected in Kmt2a mutant mice (Fig 3A). In contrast, no significant overlap was seen when genes affected in CamkIIδc TG and Kmt2b mice were compared (Fig 3A). These findings are in line with the observation that Kmt2a but not Kmt2b is reduced in CamkIIδc TG mice and further supports the idea that changes in H3K4 methylation may contribute to decreased neuronal gene expression in CamkIIδc TG mice. To test this possibility directly, we decided to measure neuronal H3K4me3 in the hippocampal CA1 region of CamkIIδc TG and control mice via

chromatin immunoprecipitation followed by next-generation sequencing (ChIP-seq). Tissue of the hippocampal CA1 region was processed and subjected to FACS to isolate neuronal nuclei using an established protocol (Fig 3B) (Benito *et al*, 2015; Halder *et al*, 2016). Afterward, H3K4me3 ChIP-seq was performed. We detected a total of 138026 H3K4me3 peaks across the entire genome. In line with previous findings from neuronal nuclei (Kerimoglu *et al*, 2017) and other tissues, the transcription start site (TSS) of genes was the major regulatory region where these peaks were localized (Fig 3C).

When we compared H3K4me3 at the TSS of CamkIIδc TG and control mice, we observed 4,627 genes with decreased and 609 genes with significantly increased H3K4me3 peaks at the TSS (Fig 3 D). It is important to reiterate that the ChIP-seq data stems specifically from neuronal nuclei of the hippocampal CA1 region allowing us to test the hypothesis that reduced neuronal H3K4me3 would explain the decreased expression of neuronal genes. Indeed, genes down-regulated in CamkIIδc TG (See Fig 1F and G) showed significantly reduced H3K4me3 level at their TSS (Fig 3E). In sum, these

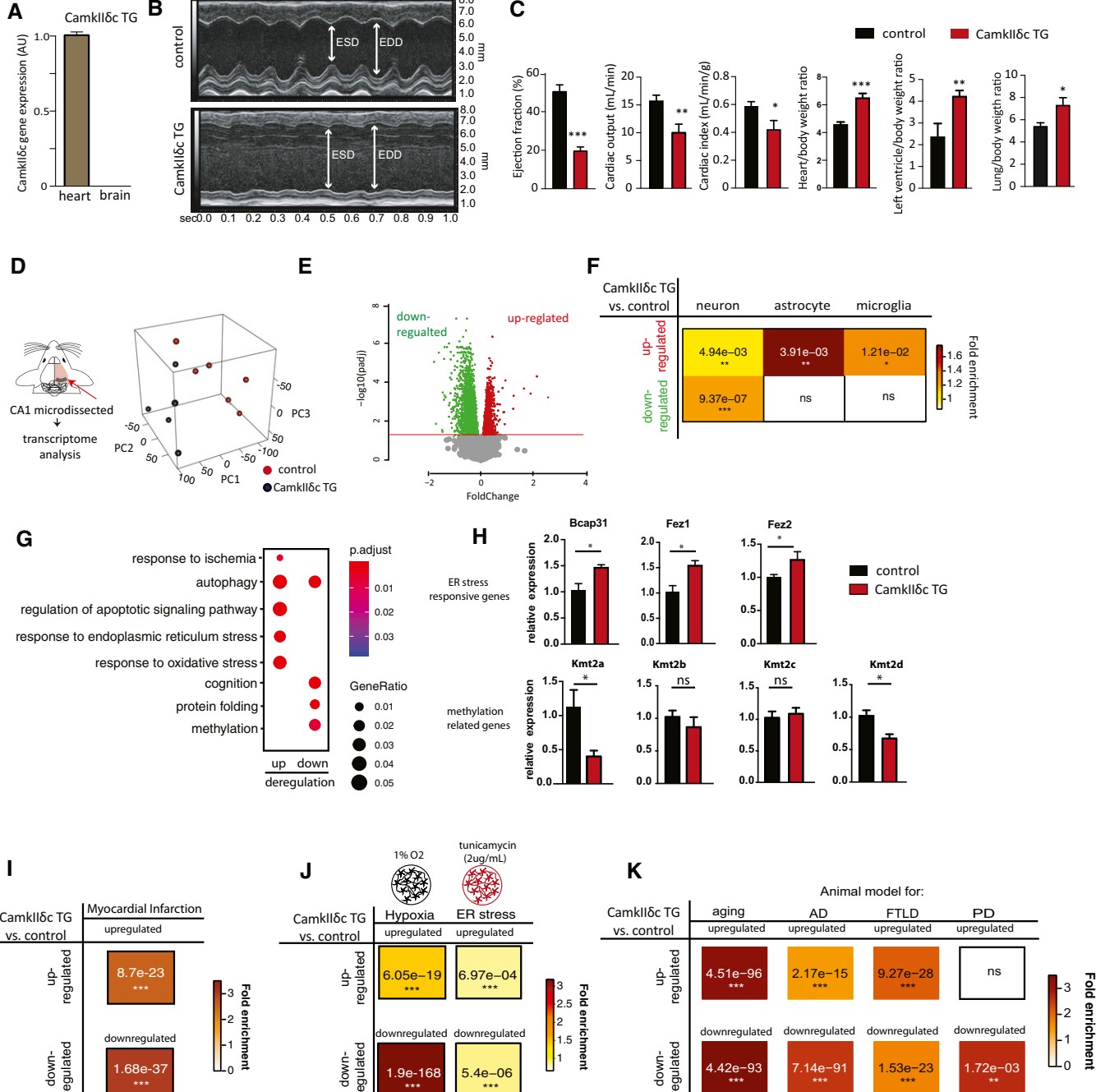

**Figure 1.**

**Figure 1.  Heart failure in CamkIIδc TG mice is linked to aberrant hippocampal gene expression.**

A   qPCR data comparing the expression of the CamkIIδc transgene in the brain and heart of 3-month-old CamkIIδc TG mice; *n* = 4/group; *\*P* < 0.05, Unpaired *t*-test, two-tailed. Note that the transgene is not detected in the brain.

B   Representative M-mode images from left ventricle from CamkIIδc TG and control mice. EDD, left ventricle diastolic diameter; ESD, left ventricle end-systolic diameter.

C   Ejection fraction, cardiac output, and index are significantly decreased in CamkIIδc TG mice (*n* = 8) when compared to control mice (*n* = 5; *\*P* < 0.05, *\*\*P* < 0.01, *\*\*\*P* < 0.001, unpaired *t*-test, two-tailed). Heart to body ratio, left ventricle to body weight ratio, and lung to body weight ratio are increased in CamkIIδc TG (*n* = 8) compared to control (*n* = 5; *\*P* < 0.05, *\*\*P* < 0.01, unpaired *t*-test; two-tailed).

D   Experimental scheme for RNA-seq analysis that was performed from hippocampal CA1 region of CamkIIδc TG mice (*n* = 6) and control mice (*n* = 5) at 3 months of age. Right panel shows principal component analysis (PCA) of the gene expression data. The first principal component (PC1) can explain 42% of the variation between two groups.

E   Volcano plot showing differentially expressed genes (FDR < 0.05). Red color indicates up-regulation, while green represents down-regulation of transcripts.

F   Hypergeometric overlap analysis comparing genes deregulated in CamkIIδc TG mice to genes uniquely expressed in neurons, astrocytes, or microglia. Numbers represent the *P* value after multiple adjustments with Benjamini–Hochberg (BH) method. Fisher's hypergeometric test; color represents fold enrichment.

G   Dot plot showing Top GO biological processes after removing redundant GO terms using Rivago.

H   qPCR quantification of selected genes reflecting ER stress or protein methylation-related processes in CamkIIδc TG mice (*n* = 4) and control mice (*n* = 5). *\*P* < 0.05, ns, non-significant, unpaired *t*-test; two-tailed. Data are normalized to Hprt1 expression.

I–K   Hypergeometric overlap analysis comparing genes deregulated in CamkIIδc TG mice to genes deregulated under (I) hypoxia conditions and (J) in response to tunicamycin-induced ER stress and (K) in hippocampal tissue from animal models of memory impairment and neurodegeneration. Benjamini–Hochberg (BH) adjusted *P* values after are denoted as numbers, and fold enrichment is represented as color heatmap. Fisher's hypergeometric test.

Data information: Bars and error bars indicate average ± standard error mean.

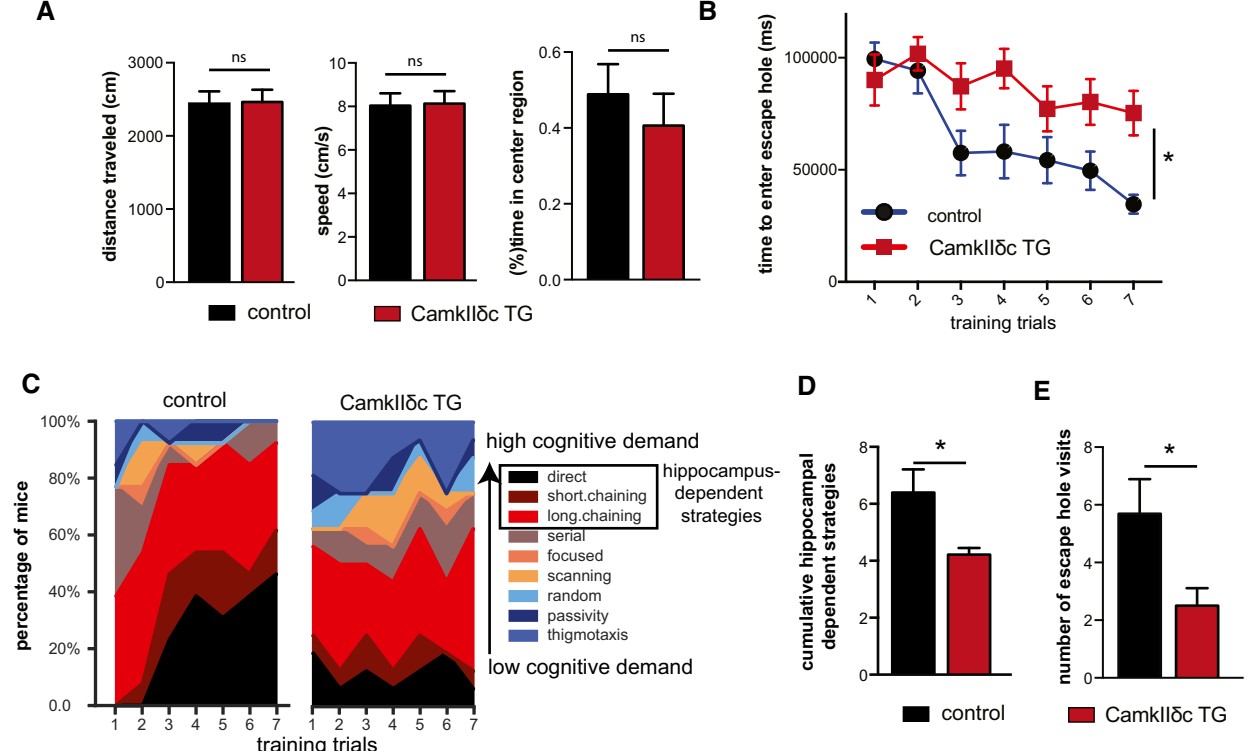

**Figure 2.  CamkIIδc TG mice display impaired hippocampus-dependent memory function.**

A   The distance traveled (left panel), the speed (middle panel), and the time spent in the center region (right panel) during a 5-min open field test was similar among 3-month-old CamkIIδc TG (*n* = 16) and control mice (*n* = 13). Unpaired *t*-test, two-tailed.

B   The time to enter the escape hole during training sessions of the Barnes maze test is impaired in old CamkIIδc TG (*n* = 16) and control mice (*n* = 13; two-way ANOVA, *\*P* < 0.05).

C   Plots showing the different search strategies of CamkIIδc TG (*n* = 16) and control mice (*n* = 13) across training trials. Each strategy is labeled with a unique color.

D   The cumulative score of hippocampus-dependent search strategies during Barnes maze training is impaired in CamkIIδc TG (*n* = 16) when compared to control mice (*n* = 13; two-tailed, unpaired *t*-test, *\*P* < 0.05).

E   Number of visits to escape hole during probe test to assay memory retrieval was impaired in CamkIIδc TG (*n* = 16) when compared to control mice (*n* = 13; two-tailed, unpaired *t*-test, *\*P* < 0.05).

Data information: Bars and error bar indicates mean ± SEM.

data provide strong evidence for the view that reduced neuronal H3K4me3 plays a crucial role in impaired neuronal gene expression observed in CamkIIδc TG mice and thereby contributes to HF induced memory loss.

## Reinstating hippocampal gene expression rescues memory impairment in CamkIIδc TG mice

Our findings point to the possibility that therapeutic strategies to increase H3K4me3 may help to ameliorate cognitive impairment in CamkIIδc TG mice and could provide a novel approach to manage cognitive impairments in HF patients. H3K4me3 is a chromatin mark linked to active gene expression and euchromatin conformation. While the inhibition of the corresponding demethylases could be one promising strategy, corresponding pharmacological compounds that would be suitable for oral administration to test the effect on cognitive function are not readily available. However, there are ample data on another group of epigenetic drugs in the context of cognitive diseases, namely histone deacetylase (HDAC) inhibitors (Fischer, 2014b). HDAC inhibitors increase histone acetylation and thereby generally favor euchromatin formation. Importantly, HDAC inhibitors were also found to affect H3K4 levels (Wang *et al*, 2009) and could, for example, reinstate impaired hippocampal H3K4me3 in mice that lack the histone methyltransferase Kmt2d (Bjornsson *et al*, 2014). Moreover, the HDAC inhibitor Vorinostat is currently tested as therapeutic intervention in AD patients (https://clinicaltrials.gov/ct2/show/NCT03056495) making it an interesting target for translational research. On this basis, we hypothesized that administration of Vorinostat would be a suitable therapeutic approach for reinstating memory function in CamkIIδc TG mice. In a pilot experiment, we found that Vorinostat was able to significantly enhance H3K9 acetylation and H3K4me3—two euchromatin marks that are functionally related (Kerimoglu *et al*, 2013; Stilling *et al*, 2014; Kerimoglu *et al*, 2017)—when administered to primary hippocampal neurons (Appendix Fig S1). Thus, 2-month-old CamkIIδc TG mice were treated orally with Vorinostat for 1 month before behavioral testing using an established protocol (Benito *et al*, 2015). Another group of CamkIIδc TG mice received corresponding vehicle solution. Vehicle-treated wild-type littermates served as additional control group (Fig 4A). All groups performed similarly in the open field test confirming our previous observation that CamkIIδc TG mice exhibit normal basal anxiety levels and motor function (Fig 4B). Moreover, Vorinostat had no effect on these parameters. Next, mice were subjected to the Barnes Maze paradigm to evaluate spatial reference memory. Consistent with our previous observation, vehicle-treated mice displayed impaired learning behavior when compared to the corresponding wild-type group (Fig 4C). In contrast, CamkIIδc TG mice treated with Vorinostat were able to master the Barnes Maze task similar to the wild-type control group (Fig 4C). Essentially, the escape latency in Vorinostat-treated CamkIIδc TG and wild-type control groups was not significantly different, suggesting that Vorinostat administration reinstates hippocampus-dependent memory function in CamkIIδc TG mice (Fig 4C). A more detailed analysis of the training procedure revealed that similar to wild-type mice, Vorinostat-treated CamkIIδc TG mice eventually adopt cognitive strategies such as direct, short, and long chaining strategies, while vehicle-treated CamkIIδc TG failed to do so (Fig 4D). Consistently, the cumulative cognitive score was

significantly impaired in vehicle-treated CamkIIδc TG mice, when compared to the wild-type control group, while no such difference was observed for Vorinostat-treated CamkIIδc TG (Fig 4E). This observation was even more obvious when mice were subjected to the memory test after seven training trials (Fig 4F). These data show that oral administration of Vorinostat ameliorates learning and memory impairment in CamkIIδc TG mice.

## Vorinostat ameliorates gene expression changes in CamkIIδc TG mice

Vorinostat treatment of CamkIIδc TG mice had no significant effect on cardiac pathology (Fig EV2) suggesting that reinstatement of memory function in our experimental system is most likely linked to brain-specific processes. Thus, we analyzed gene expression in the hippocampal CA1 region of vehicle-treated wild-type mice as well as in vehicle and Vorinostat-treated CamkIIδc TG mice via RNA-seq (Fig 5A). In line with our previous observation (See Fig 1D–G), RNA-seq data analysis revealed a major deregulation of gene expression in vehicle-treated CamkIIδc TG mice compared to the vehicle-treated wild-type control group (Fig EV3A and B). Our further analysis shows that Vorinostat could partially restore physiological gene expression in CamkIIδc TG mice (Fig EV3B and C). The finding that Vorinostat treatment increases the expression of genes that were down-regulated in CamkIIδc TG mice can easily be explained by the effect of Vorinostat on euchromatin formation. However, the observation that Vorinostat also decreases the expression of genes that were elevated in CamkIIδc TG mice is most likely due to additional mechanisms.

To further elucidate this, we decided to investigate the RNA-seq data in greater detail. Recent studies showed that the detection of regulatory co-expression modules is a suitable approach to further understand transcriptional plasticity in health and disease (Gandal *et al*, 2018). To this end, we performed weighted gene co-expression analysis (Langfelder & Horvath, 2008) (Fig 5B) and identified 14 different modules in the entire RNA-seq dataset (see Materials and Methods for details, Figs 5C and, EV3D and E). Two of these modules—namely RNA module 1 and 2—exhibited significantly different expression among vehicle-treated CamkIIδc TG and wild-type control mice. RNA module 1 was decreased in vehicle-treated CamkIIδc TG mice, while its expression was partially rescued upon Vorinostat treatment (Fig 5C). Gene ontology analysis suggested that the genes of RNA module 1 are linked to cognition, learning, and memory (Fig 5C). Further analysis identified a cluster of 30 hub genes within module 1. Notably, 26 of these genes were shown to cause to memory impairment when their expression was manipulated (Fig 5D; Dataset EV5). Interestingly, decreased expression of RNA module 1 was also observed in MI mice (Fig EV4A), further supporting our data that similar hippocampal gene expression changes are observed in the two HF models. Partial gene expression of RNA module 1 after Vorinostat treatment may be explained by its effect of chromatin that likely involves enhanced H3K4me3. To provide evidence for this hypothesis, we performed H3K4me3 ChIP-seq from sorted hippocampal neurons of CamkIIδc TG that were either treated with vehicle or Vorinostat. Vehicle-treated wild-type mice served as additional control. We were able to reproduce our initial findings and observed that that H3K4me3 is decreased at the TSS of the genes within RNA module 1 in hippocampal neurons of

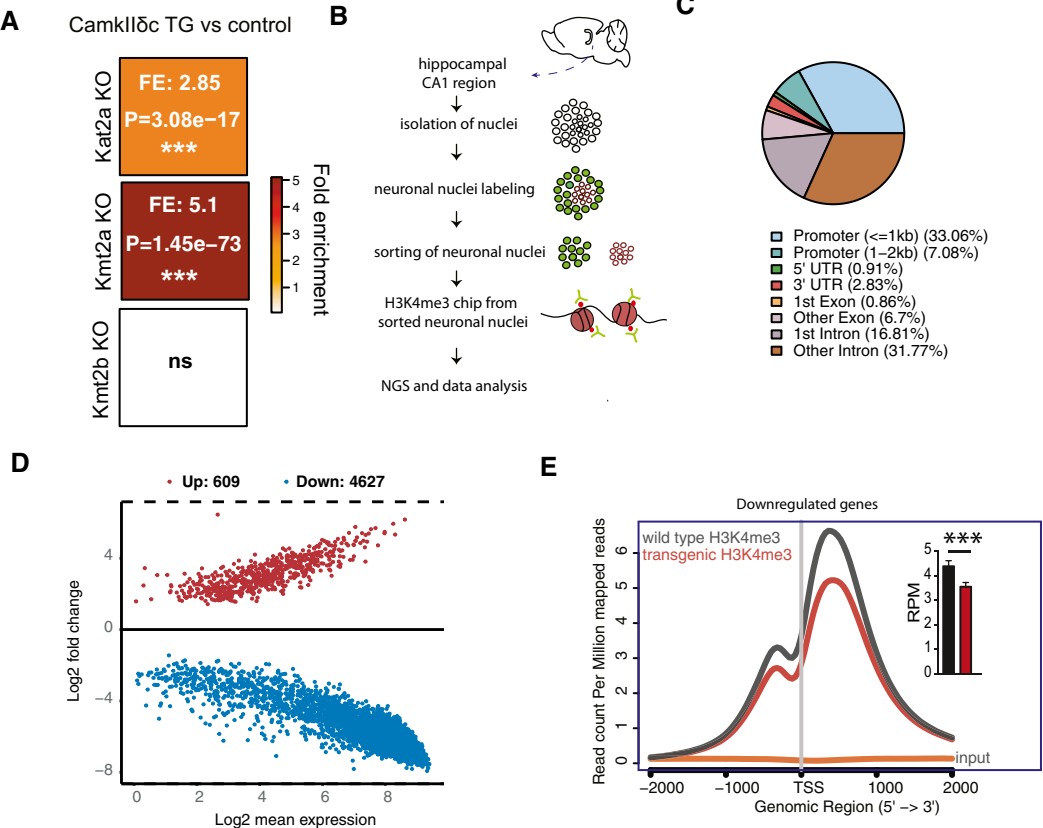

**Figure 3. Neuronal H3K4m3 is impaired in the hippocampus of CamkIIδc TG mice.**

A  Hypergeometric overlap analysis comparing genes deregulated in CamkIIδc TG mice to genes differentially expressed in the hippocampal CA1 region of Kat2A, Kmt2A, and Kmt2b knockout mice. Fisher's hypergeometric test, Benjamini–Hochberg (BH) correction.

B  Experimental scheme for ChIP-seq analysis. NeuN + neuronal nuclei were FACS sorted and used to perform chromatin immunoprecipitation (ChIP) for H3K4me3.

C  Pie chart showing the distribution of H3K4me3 peaks in the neurons of the hippocampal CA1 region from CamkIIδc TG mice.

D  MA plot showing the number of significantly altered neuronal H3K4me3 peaks when comparing CamkIIδc TG and control mice.

E  NGS plot showing H3K4me3 peaks at the TSS of genes down-regulated in the hippocampus of CamkIIδc TG mice. Inset shows statistical analysis, (unpaired t-test, two-tailed, ***P < 0.001). Bars and error bars indicate mean ± SEM. H3K4me3 ChIP analysis was performed on four replicates/group.

vehicle-treated CamkIIδc TG when compared to vehicle-treated wild-type control littermates (Fig 5E). Importantly, we observe that Vorinostat treatment reinstated H3K4me3 across the TSS of the genes within RNA module 1 in CamkIIδc TG mice thereby providing a feasible explanation for the restored expression of the "learning and memory genes" of the RNA module 1 that is decreased in hippocampal CamkIIδc TG mice (Fig 5E).

In contrast, RNA module 2 was significantly increased in vehicle-treated CamkIIδc TG mice when compared to the vehicle-treated wild-type control group (Fig 5F). Increased expression of RNA module 2 was also observed in MI mice (Fig EV4B). Expression of this cluster was partially decreased to control levels in Vorinostat-treated CamkIIδc TG mice (Fig 5F). In line with our previous analysis of up-regulated genes in CamkIIδc TG mice, the genes of RNA module 2 were mainly linked to cellular stress-related pathways (Fig 5F and G) and showed a significant overlap to genes increased in response to hypoxia in neuronal cultures (this study), human brain organoids exposed to hypoxia (Pa ca et al, 2019), or ER stress (Appendix Fig S2). However, the question remained how Vorinostat, an epigenetic drug that is linked to euchromatin formation and

the activation of gene expression, would decrease the observed pathological gene expression response linked to hypoxia and cellular stress pathways. One possible explanation is that Vorinostat induces molecular processes that antagonize this type of pathological gene expression. Several candidate mechanisms likely play a role; however, the regulation of microRNAs appeared to us as a promising hypothesis to be tested first. MicroRNAs are small noncoding RNAs that regulate cellular homeostasis via binding to a target mRNA thereby causing its degradation or inhibition of translation (Gurtan & Sharp, 2013). Compensatory microRNA responses have been described in response to various cellular stress conditions (Kagias et al, 2012), and previous studies showed that Vorinostat treatment affects microRNA expression in disease models thereby contributing to the therapeutic effect (Ali et al, 2015; Benito et al, 2015; Ye et al, 2018). Thus, we hypothesized that Vorinostat-induced microRNA expression might help to explain—at least in part—the effect on RNA module 2 in CamkIIδc TG mice. To this end, we performed small RNA sequencing of the hippocampal CA1 region obtained from control and vehicle-treated CamkIIδc TG mice as well as from Vorinostat-treated CamkIIδc TG mice. Differential

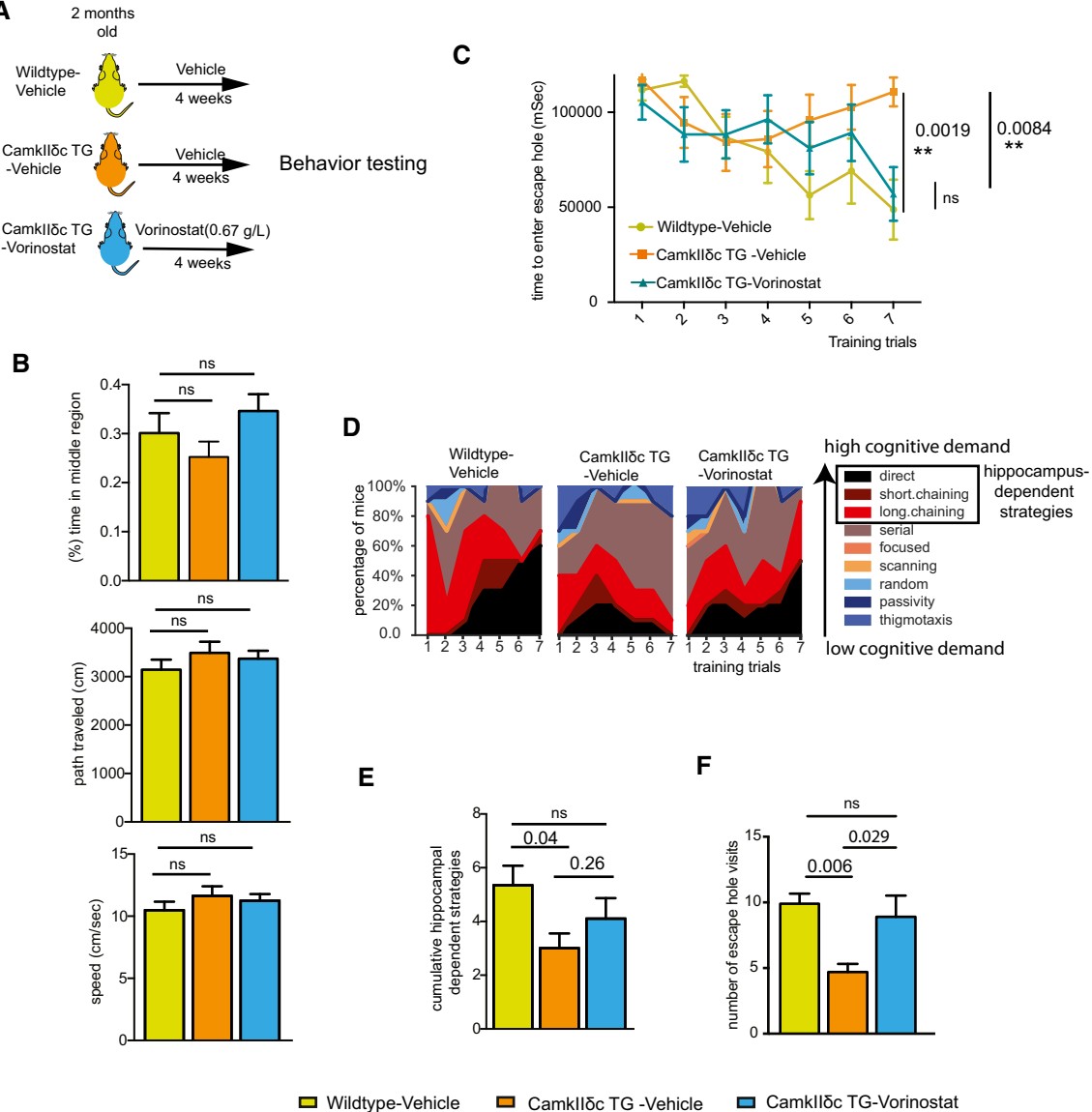

**Figure 4. Vorinostat reinstates memory function in CamkIIδc TG mice.**

A  A schematic outline of the experimental design.

B  The distance traveled (upper panel), the speed (middle panel), and the time spent in the center region (lower panel) during a 5-min open field test were similar among groups (*n* = 10/group). One-way ANOVA.

C  Latency to enter the escape hole during Barnes maze training (two-way ANOVA). *n* = 10 per group.

D  Plots showing the different search strategies across training trials. Each strategy is labeled with a unique color.

E  Cumulative hippocampus-dependent strategy scores during the Barnes maze training (one-way ANOVA). *n* = 10 per group.

F  Number of visits to the escape hole during probe test (*n* = 10/group, one-way ANOVA).

Data information: Bars and error bars indicate mean ± SEM.

expression analysis revealed a number of regulated microRNAs when comparing the various conditions (Dataset EV6). To specifically identify microRNA networks that could explain the decreased expression of cellular stress response genes upon Vorinostat treatment, we performed a weighted co-expression analysis (Langfelder & Horvath, 2008) (Fig 5H) and identified five microRNA modules (Fig EV5A and B). One module—namely microRNA module 2—was significantly decreased in vehicle-treated CamkIIδc TG mice when

compared to the vehicle control group (Fig 5H), while its expression was increased to physiological levels upon Vorinostat treatment (Fig 5H). Next, we asked whether the increased expression of microRNA module 2 would be correlated with the corresponding expression of stress response genes increased in the hippocampal CA1 region of CamkIIδc TG mice. To this end, we first performed a pairwise correlation analysis between genes and microRNAs that were differentially expressed in Vorinostat-treated vs. vehicle-

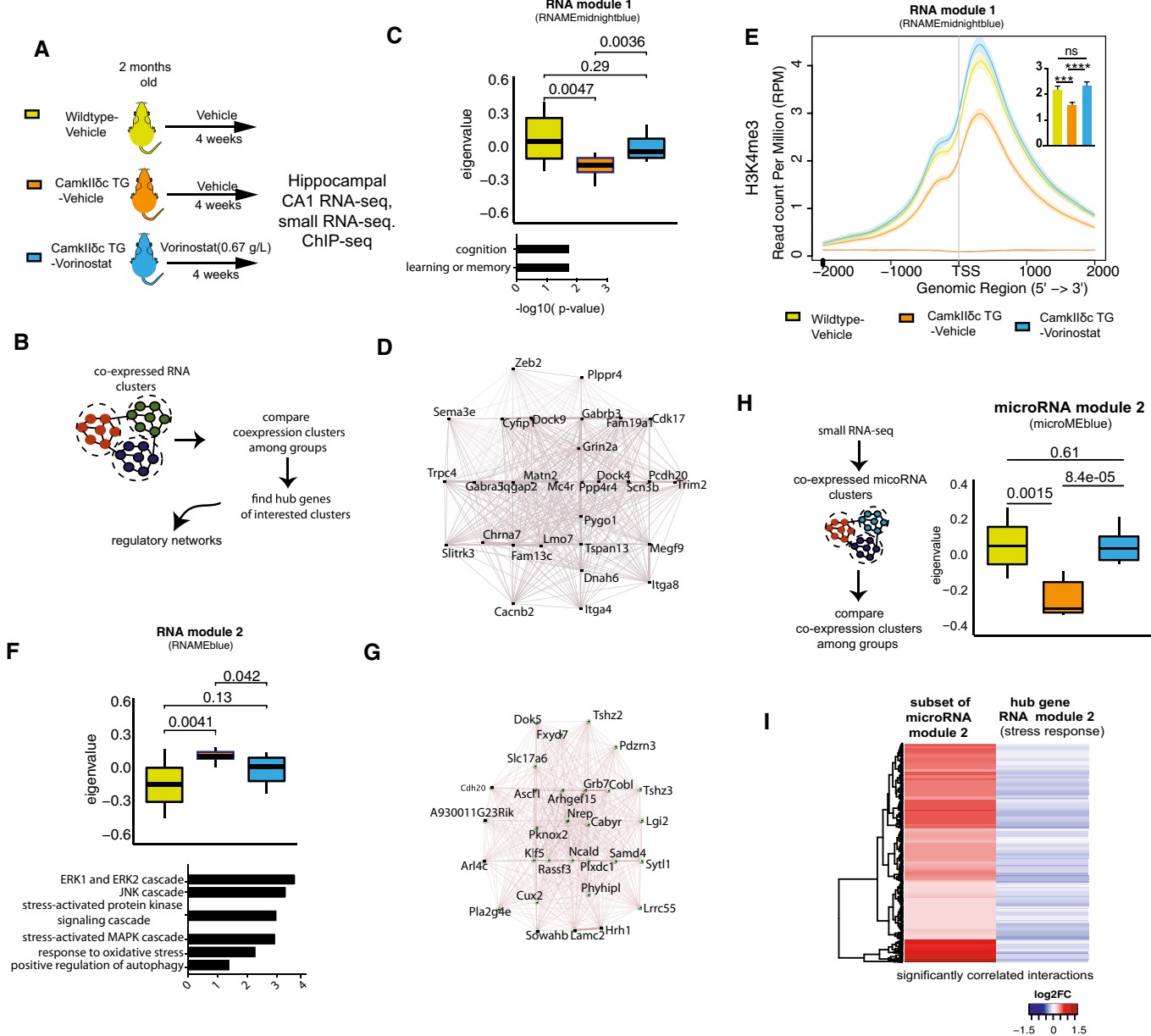

**Figure 5. Vorinostat ameliorates pathological hippocampal gene expression in CamkIIδc TG mice.**

A   Schematic outline of the experimental design.

B   Scheme for WGCNA analysis.

C   Upper panel: Expression of RNA module 1 among the three experimental groups. Kruskal–Wallis test. Lower panel: Gene ontology analysis of genes that are part of RNA module 1.

D   Network representing top 30 hub genes of the gene network based on RNA module 1.

E   H3K4me3 at TSS region of genes from RNA module 1. Inset shows statistical comparisons among groups (one-way ANOVA, ***$P < 0.001$, ****$P < 0.0001$). Bars and error bars indicate mean ± SEM. Number of replicates analyzed for H3K4me3 profile at TSS (wild-type vehicle: 3; CamkIIδc TG vehicle $n = 4$; CamkIIδc TG-Vorinostat $n = 4$).

F   Upper panel: RNA module 2 and its expression among the three experimental groups. Kruskal–Wallis test. Lower panel: Functional annotations of the genes that are part of RNA module 2. Wild-type vehicle: $n = 10$; CamkIIδc TG vehicle $n = 9$; CamkIIδc TG-Vorinostat $n = 9$.

G   Gene correlation network of the top hub genes ($n = 30$) of RNA module 2.

H   Left panel: Schematic outline of the analysis of microRNA-sequencing data. Right panel: Expression of microRNA module 2 among experimental groups. Kruskal–Wallis test. Wild-type vehicle: $n = 9$; CamkIIδc TG vehicle $n = 7$; CamkIIδc TG-Vorinostat $n = 10$.

I   Heatmap showing significant negative correlation (FDR < 0.05) between microRNA members of microRNA module 2 and hub genes from RNA module 2 (see panel G).

Data information: In the boxplots in (C, F, and H), the center line indicates the median, while the upper and lower lines represent the $75^{th}$ and $25^{th}$ percentiles, respectively. The whiskers represent the smallest and largest values in the 1.5× interquartile range.

treated CamkIIδc TG mice. We observed that microRNAs within microRNA module 2 showed a significant negative correlation to the hub genes of RNA module 2, representing the module linked to cellular stress responses and autophagy (Fig 5I). These data suggest that Vorinostat treatment in CamkIIδc TG mice increases the expression of microRNAs that antagonize the expression of genes linked to pathological cellular stress. Further evidence for this view stems from the finding that these microRNAs are mainly encoded within genes that exhibit reduced hippocampal H3K4me3 in CamkIIδc TG mice (Fig EV5C) and display restored H3K4me3 profile after Vorinostat treatment (Fig EV5D). It is, however, likely that additional mechanisms than microRNA expression also contribute to this effect.

In sum, these data suggest that aberrant neuronal gene expression plays a central role in HF associated cognitive decline. In turn, approaches that target these gene expression changes could provide a novel therapeutic avenue to manage cognitive dysfunction in HF patients.

## Discussion

By employing a genetic mouse model for HF, we show for the first time that HF leads to substantial changes in hippocampal gene expression. The genes that were up-regulated significantly overlap with genes deregulated in neurons exposed to cellular stress such as oxidative and ER stress. These data suggest that cardiac dysfunction, which has been linked to reduced blood flow to the brain (Bikkina et al, 1994; Verdecchia et al, 2001), initiates a cellular stress response that eventually manifest at the level of neural gene expression. Support for this view stems from our observation that similar changes in hippocampal gene expression were detected in a mouse model for MI. Other models for HF, such as for example the trans-aortic constriction (TAC) model, should also be tested in the future. Cerebral hypoperfusion is, however, only one possible upstream mechanisms, and other processes are likely equally important. For example, there is clear evidence that in addition to hormonal regulation, other circulating factors such as small noncoding RNA or proteins regulate biological process in an inter-organ-dependent manner that could affect brain function (Jose, 2015; Pluvinage & Wyss-Coray, 2020). Our results also reveal that these hippocampal gene expression changes in mice suffering from HF parallel the changes observed in models for neurodegenerative diseases (Benito et al, 2015; Gispert et al, 2015; Gjoneska et al, 2015; Swarup et al, 2018). This is in line with previous reports suggesting that hypoxia-mediated oxidative and ER stress are early and common events in neurodegenerative diseases that can trigger subsequent pathological changes associated with memory loss (Feldstein, 2012; Xiang et al, 2017; Butterfield & Halliwell, 2019). Indeed, further analysis of the data revealed that the hippocampal genes down-regulated in response to HF represent cellular processes linked to cognition and are similar to the gene expression changes observed in models for dementia. These findings suggest that activation of cellular stress pathways might be one reason for the down-regulation of hippocampal genes essential for cognition. Support for this view stems from our observation that the sole exposure of neuronal cultures to hypoxia or ER stress leads to the down-regulation of such neuronal gene-sets. In line

with these gene expression data, we show that CamkIIδc TG mice exhibit impaired hippocampus-dependent learning and memory. Although our report of memory impairment in a HF mouse model is novel, these data are in agreement with various studies in humans showing that cardiac dysfunction is associated with cognitive decline and an increased dementia risk (Angermann et al, 2012; Ampadu & Morley, 2015; Doehner et al, 2017). Furthermore, memory impairment has been reported in animal models for acute myocardial ischemia (Evonuk et al, 2017) and various models for chronic cerebral hypoperfusion but the underlying molecular mechanisms remained poorly understood so far (e.g. see (Patel et al, 2017)). It is interesting to note that HF has also been associated with depression, and several studies have addressed this issue for example in rodent models for MI. The corresponding data suggest, however, that MI can lead to either an acute or late onset of depressive-like phenotypes (Frey et al, 2014; Bruns et al, 2019). In our experimental setting, neither CamkIIδc TG (unpaired t-test; $P = 0.39$, $n = 7$/group) nor MI mice (unpaired t-test; $P = 0.1$, $n = 10$ sham control group; $n = 8$, MI group) exhibited significantly altered depressive-like behavior when tested in the porsolt forced swim test (Agís-Balboa et al, 2017). Since depression is a well-known comorbidity with cognitive decline and its pathogenesis has been linked to epigenetic gene expression (Alexopoulus, 2019), it will be important to further explore the potential mechanistic links HF, depression, and dementia in future studies. How precisely activation of the various cellular stress pathways leads to the down-regulation of genes essential for cognition remains to be investigated and is likely to be multifactorial making it difficult to identify suitable targets for therapeutic intervention. From a therapeutic point of view, the fact that hippocampal genes linked to cognition are eventually decreased might offer a more promising avenue to treat cognitive defects in HF patients, especially since these patients usually already suffer from the disease for a prolonged time period. In this context, it is important to reiterate that our data suggest that HF eventually leads to the down-regulation of gene clusters important for cognition via processes linked to reduced histone-methylation, especially deceased levels of the euchromatin mark H3K4me3. These findings are in line with current literature showing that proper neuronal H3K4me3 is essential for memory consolidation (Gupta et al, 2010; Kerimoglu et al, 2013; Jakovcevski et al, 2015; Kerimoglu et al, 2017). Our data hint at a specific role of the H3K4 methyltransferase Kmt2a, which is down-regulated in the hippocampus of CamkIIδc TG mice. These findings are in line with recent reports showing that mice lacking Kmt2a in excitatory neurons of the hippocampus exhibit impaired learning and memory and decreased expression of genes implicated in cognitive function (Kerimoglu et al, 2017). Indeed, our data show that genes deregulated in the hippocampi of Kmt2a knockout mice—but not of Kmt2b—significantly overlap with deregulated genes in CamkIIδc TG mice. While H3K4me3 is an important euchromatin mark, epigenetic gene expression control is mediated by variable combinations of histone modifications (Wang et al, 2008) and changes of other histone modifications likely play a role, a question that should be addressed in future research. Taken together, these data point to a scenario in which HF leads to hypoxia and cellular stress, eventually driving loss of neuronal euchromatin causing decreased expression of neuronal plasticity genes essential for cognition (Fig 6). Further support for this view

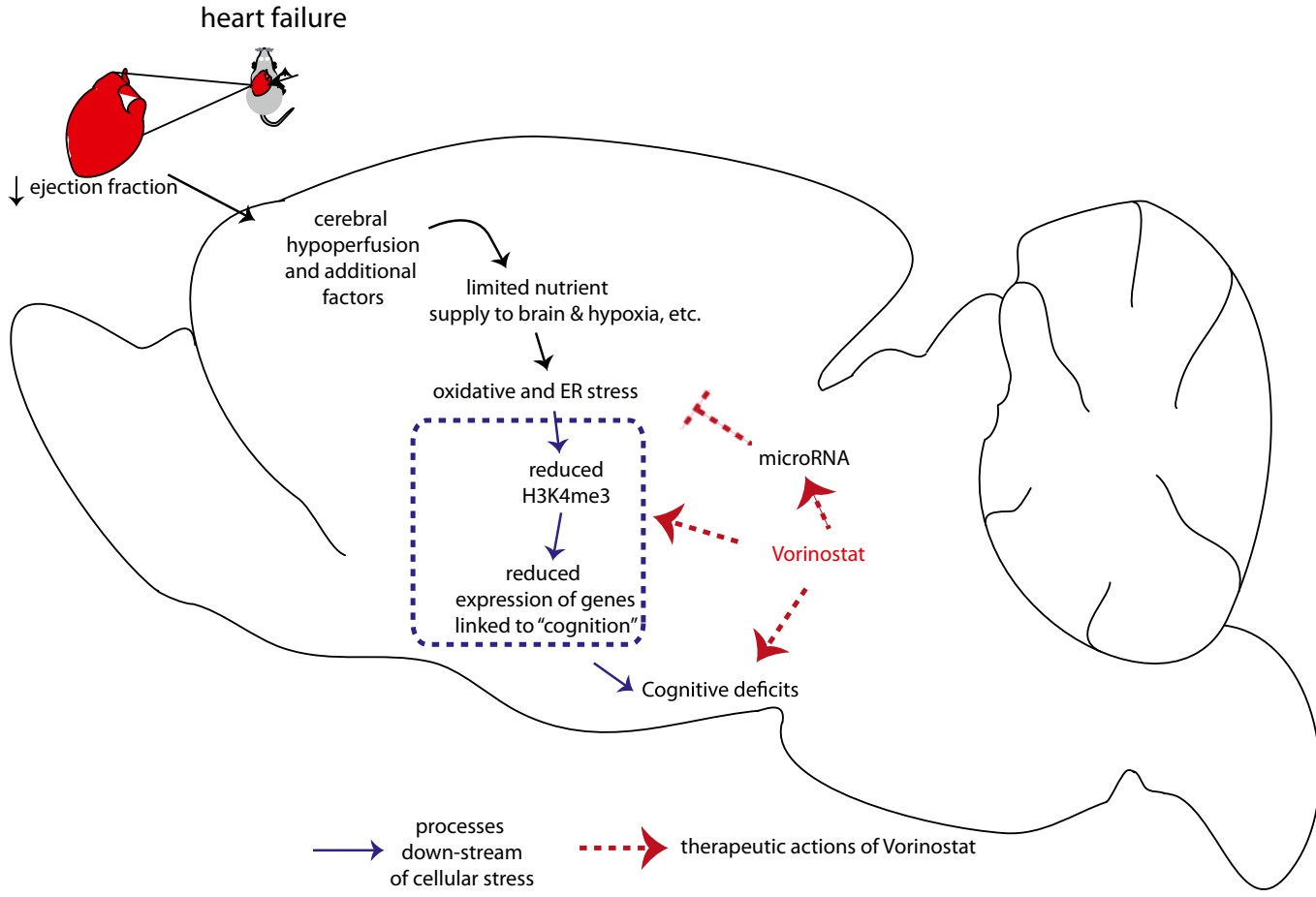

**Figure 6. Model summarizing how heart failure contributes to memory impairment and corresponding option for therapeutic intervention.**

Cardiac insufficiency leads to cerebral hypoperfusion, which is in line with a hippocampal gene expression response linked to oxidative, and ER stress. It is likely that other factors in addition to hypoperfusion, such as circulating factors that may affect brain function, also play a role. Our data suggest that oxidative and ER stress drive reduced expression of genes important for memory function, which involves reduced neuronal H3K4me3 representing loss of euchromatin. Other epigenetic changes are likely to further contribute to this effect. Administration of the HDAC inhibitor Vorinostat partially increases the expression of memory-related genes but also decreases the expression of genes linked to oxidative and ER stress via the induction of a compensatory microRNA cluster.

stems from our data that administration of an epigenetic drugs that promotes euchromatin formation reinstates memory in CamkIIδc TG mice and that this effect cannot be simply explained by improved cardiac output. These findings pave the road to a novel therapeutic approach to treat HF-induced cognitive dysfunction and lower the risk for age-associated dementia in these patients. To this end, although cerebral hypoperfusion and cellular stress appear to be initial events in the development of cognitive decline in patients suffering for cardiac dysfunction, our data suggest that they eventually lead to epigenetic changes that involve altered H3K4me3 but most likely also other histone modifications in neurons. Such epigenetic alterations are known to represent long-term adaptive changes that can persist even in the absence of the initial stimulus (Fischer, 2014a). Thus, targeting the epigenome has emerged as a promising therapeutic option to treat complex and multifactorial diseases including dementia, even at an advanced stage of the disease (Fischer, 2014a; Fischer, 2014b). In fact, previous studies showed that other risk factors for dementia

such as aging (Peleg et al, 2010; Benito et al, 2015), protein aggregation, (Kilgore et al, 2010; Govindarajan et al, 2011; Benito et al, 2015; Gjoneska et al, 2015), neuropsychiatric diseases (Nestler et al, 2015), or peripheral inflammation (Wendeln et al, 2018) lead to similar changes representing a loss of neuronal euchromatin and reduced expression of genes linked to cognition. Of note, therapeutic strategies toreinstate euchromatin related gene expression were able to improve memory function in such models (Benito et al, 2015; Bahari-Javan et al, 2017). For example, inhibitors of HDAC have emerged as promising candidates to treat cognitive decline, and the FDA approved HDAC inhibitor Vorinostat is currently undergoing trials in Alzheimer's disease patients (Clinica lTrials.gov Identifier: NCT03056495). As mentioned above, oral administration of Vorinostat to CamkIIδc TG mice improved their learning and memory abilities. These findings cannot be explained by improved cardiac function, since a 1-month treatment of CamkIIδc TG mice with Vorinostat had no significant effect on cardiac pathology in our experimental setting. It is important to

note that HDAC inhibitors have been suggested as drugs to treat cardiac diseases (McKinsey, 2012), and Vorinostat was found to improve cardiac function, for example, in models for HF with preserved ejection fraction (Wallner et al, 2018) or MI (Nagata et al, 2019). However, either higher concentrations or longer treatment durations were applied in these studies. It is nevertheless important to note that we observed a non-significant trend for reduced hypotrophy in Vorinostat-treated CamkIIδc TG mice, and it will be interesting to examine, for example, whether factors secreted from corresponding cardiac tissue would modulate brain function. Our data show that Vorinostat treatment increased the expression of formerly down-regulated hippocampal genes linked to cognition. In fact, our detailed analyses revealed that Vorinostat reinstated the expression of a specific gene cluster in which nearly every hub gene was shown to be essential for memory function. Hence, reducing either of these genes alone was found to cause memory impairment (see Dataset EV5). While these findings are in line with the know role of Vorinostat to induced euchromatin and gene expression, it was surprising to see that Vorinostat-treated CamkIIδc TG also exhibited reduced expression of genes linked to cellular stress responses. Our data suggest that this effect is mediated via the induction of a compensatory microRNA network that down-regulated cellular stress response hub genes (Fig 6). Furthermore, the relevant microRNAs may also be affected by H3K4me3 but the precise mechanism of how Vorinostat regulates the levels of selected microRNAs remains to be investigated. However, these findings are in line with the reported role of the microRNAome as one key molecular process to maintain cellular homeostasis. In fact, several reports link microRNA expression to compensatory mechanisms in various diseases and Vorinostat may favor the induction of such mechanisms (Gebert & MacRae, 2019). We cannot exclude the possibility that the improved memory function in response to Vorinostat treatment is mediated by other mechanisms than the ones described in our study. For example, Vorinostat also acts on non-histone proteins and has been found to suppress hypoxia signaling in cancer models (Zhang et al, 2017). Moreover, although Vorinostat increases H3K4me3, this indirect effect is most likely mediated by increased histone acetylation that generally promotes euchromatin formation. In line with this, previous findings show that Kmts act in concert with histone acetyltransferases (Kerimoglu et al, 2013; Kerimoglu et al, 2017; Husmann & Gozani, 2019). Nevertheless, in the future, it will be important to investigate whether therapeutic approaches that target more directly H3K4me3 would be even more efficient to reinstate memory function in response to HF. Suitable strategies could be the inhibition of corresponding demethylases or activation of methyltransferases. The knowledge about these enzymes in the adult brain is, however, only emerging and safe drugs that could be tested for oral application in the context of cognitive dysfunction are not available yet.

In conclusion, our data elucidate the molecular mechanisms by which cardiac dysfunction contributes to cognitive impairment and suggest a key role for epigenetic neuronal gene expression. Targeting gene expression changes in the brain, through drugs such as HDAC inhibitor Vorinostat, ameliorate memory impairment and partially reinstate physiological gene expression. Thus, therapeutic strategies that target epigenetic gene expression may be a suitable approach to treat cognitive dysfunction even in chronic HF patients

and lower their risk of developing age-associated cognitive diseases such as AD.

## Material and Methods

### Animals and tissue preparation

CamkIIδc transgenic and wild-type littermates were housed in standard cages on 12 h/12 h light/dark cycle with food and water ad libitum. All experimental procedures were approved by a local animal care committee of the University Medical Center, Goettingen, Germany and the responsible Lower Saxony State office. The genetic background of all mice was C57Bl/6J. Three-month-old mice were analyzed for the experiments. The rational for choosing this time point was based on previous findings showing that at 3 months of age, wild-type mice display good cognitive performance (Peleg et al, 2010), while at 3 month of age CamkIIδc transgenic mice exhibit robust dilated HF and altered myocyte $Ca^{2+}$ regulation (Sossalla et al, 2011). MI was introduced via an established protocol in our laboratory through coronary artery ligation Sham-operated mice (similar surgery, but no artery ligation) was used as controls (Mohamed et al, 2018). Vorinostat treatment was performed following our detail protocol published previously (Benito et al, 2015). Male mice were used in the experiments. For tissue preparation, animals were sacrificed by cervical dislocation. Hippocampal subregion CA1 was isolated, snap-frozen in liquid nitrogen, and stored at −80°C. Hearts were dissected by a cut above the base of the aorta and perfused with 0.9% sodium chloride solution until blood free, snap-frozen in liquid nitrogen, and stored at −80°C. In addition, lung and tibia were extracted and their respective weight or length was determined.

### Echocardiography

The heart function and dimensions were examined by echocardiography using a Vevo 2100 imaging platform (Visualsonics) with 30MHz transducer (MS-400). The animals were anesthetized with isoflurane (1–2%) and M-mode sequences of the beating heart recorded in the short axis and the long axis, respectively. The images were used to determine the left ventricular end-diastolic and end-systolic volumes (area*length*5/6). These parameters were used to calculate the ejection fraction as indicator of left ventricular heart function. The investigator was blinded to genotype and age.

### Mouse behavior

Open field test was performed according to a previous study (Bahari-Javan et al, 2012). Briefly, mice were placed gently in the middle quadrant of an open field and allowed to explore the arena for 5 min. The travel trajectories were recorded using VideoMot (TSE-Systems). Barnes Maze experiment was performed according to Sunyer et al (2007) with the following modifications. Briefly, the barnes maze arena was placed in a room with eight extra high intensity light and sound system (creative) The test consisted three parts: (i) habituation, (ii) training, and (iii) test. During habituation and training, all twenty holes on the maze were closed except one with an escape cage placed underneath. During habituation on first day,

mice were placed in the middle region of the maze and lights along with buzzer in the sound system were turned on. Mice were gently nudged to escape hole and let it to explore escape hole for 2 min. After 2 min, mice were put back to original cages. Escape cage was cleaned with 70% EtOH, and bed material was changed after each trial. Training sessions were performed from day 2 to day 4, and three trials were performed each day. On day 5, probe test was performed where the escape hole was closed and time that mice were spending in escape hole quadrant, distance from the escape hole were enumerated from the video recordings using VideoMot (TSE-Systems). We adapted MAST-C (Illouz *et al*, 2016) algorithm with slight modification for Barnes maze data analysis and extracted few additional features including total coverage, distance from centroid to platform, and path efficiency from raw data. Different strategies were defined based on trajectories mice employed each trial. Cumulative score for hippocampal-dependent strategy score was calculated as follow:

$$\text{mean cumulative strategy score} = \frac{1}{n} * \sum_{i=1}^{n} \left[ \frac{\text{Sdf}_i * \text{Sdc}}{\text{mc}} + \frac{\text{Sscf}_i * \text{Sscc}}{\text{mc}} + \frac{\text{Slcf}_i * \text{Slcc}}{\text{mc}} \right]$$

$n$ = total trial number; $\text{Sdf}_i$ = frequency of direct strategy in $i^{\text{th}}$ trial; Sdc = given strategy score for direct search; 10; mc = total number of mice; $\text{Sscf}_i$ = Frequency of short chaining in $i^{\text{th}}$ trial; Sscc = given strategy score for short chaining search; 9.5; $\text{Slcf}_i$ = frequency of short chaining in $i^{\text{th}}$ trial; Slcc = given strategy score for long chaining search; 7.

### RNA isolation and sequencing

RNA isolation was performed using RNA Clean and Concentrator kit according to manufacturer protocol without modifications. Concentration was measured on nanoDrop, and quality of RNA was evaluated. For mRNA sequencing, 500 ng total RNA was used as input to prepare cDNA libraries according to Illumina Truseq and 50 bp sequencing reads were run in HiSeq 2000. For small RNA sequencing, 100 ng total RNA was used as initial input. Small RNA was enriched using size selection from based on gel. cDNA library and sequencing have been performed according to manufacturer's protocol (NEBNext Small RNA library prep set for Illumina). Next-generation sequencing was performed on HiSeq 2000 platform.

### Chromatin immunoprecipitation for H3K4me3

Chromatin immunoprecipitation was performed according to Halder *et al* (2016) with 0.2 μg chromatin and 1μg H3K4me3 (ab8580) antibody. ChIP-seq library preparation was performed using NEBNext Ultra II DNA library preparation according to manufacturer's protocol. 2 nM libraries were pooled and sequenced in Illumina Hiseq 2000 with 50-bp single end reads. Briefly, flash frozen hippocampal sub-regions CA1 were isolated and homogenized briefly in low-sucrose buffer (320 mM Sucrose, 5 mM $CaCl_2$, 5 mM $MgAc_2$, 0.1 mM EDTA, 10 mM HEPES pH 8, 0.1% Triton X-100, 1 mM DTT, supplemented with Roche protease inhibitor cocktail) with plastic pestles in 1.5-ml tubes. 1% formaldehyde was added as a final concentration and incubated at room temperature for 10 min on a rotating wheel. 125 mM glycine was added as final

concentration and incubated for 5 min at room temperature for quenching remaining formaldehyde. The homogenate was centrifuged at 2,000 $g$ for 3 min at 4°C, and supernatant was discarded. The remaining crude nuclear pellet was resuspended in additional low-sucrose buffer and further homogenized with a mechanical homogenizer (IKA Ultraturax T10, with S10N-5G tool). The solution was carefully layered onto 6 ml high-sucrose buffer (1,000 mM Sucrose, 3 mM $MgAc_2$, 10 mM HEPES pH 8, 1 mM DTT, Roche protease inhibitor) in oak-ridge tubes and centrifuged at 3,220 $g$ for 10 min at 4°C in a swinging bucket rotor centrifuge to get rid of myelin debris. The resulting nuclear pellet was resuspended into left over buffer, transferred into 2-ml microfuge tubes (DNA-low bind), and centrifuged for 3 min at 2,000 $g$ at 4°C to recover the nuclear pellet. It was resuspended into 500 μl PBTB buffer (PBS-Tween-BSA buffer, 1% BSA, 0.2% Tween-20, protease inhibitor in 1X PBS), and 0.5 μl Anti-NeuN-Alexa488 conjugated antibody (MAB377X, 1:1,000) was added. After incubating for 20 min at 4°C, samples were washed with PBTB once and proceeded with the sorting in FACSAria III with 85μm nozzle. Sorted nuclei were collected into PBTB coated falcon tubes and pelleted with brief centrifugation, and pellets were flash frozen and saved at −80°C until further processing for ChIP-seq. Only NeuN + neuronal nuclei were used to perform chromatin immunoprecipitation (ChIP). Chromatin amount normalization and ChIP was performed according to Halder *et al* (2016) with 0.2 μg chromatin and 1 μg H3K4me3 (ab8580) antibody. ChIP-seq library preparation was performed using NEBNext Ultra II DNA library preparation according to manufacturer's protocol. 2 nM libraries were pooled and sequenced in Illumina Hiseq 2000 with 50-bp single end reads.

### Modeling hypoxia and endoplasmic reticulum stress in primary neurons

Primary hippocampal neuronal culture was prepared as described previous (Benito *et al*, 2015). Briefly, primary neuronal culture was prepared from E17 embryos from CD1 pregnant mice (Janvier Labs). Mouse embryonic hippocampi were microdissected and digested with 5 ml mixture of prewarmed PBS + 2.5% trypsin (Gibco) for 13 min at 37°C. Subsequently, cells were washed two times with media 1 (DMEM, 10% FBS, 1% penicillin/streptomycin, filtered sterilized) followed by homogenization in 1 ml of media 1. Next, cells were centrifuged at 200 $g$ for 5 min (37°C) before resuspending in 1 ml of media 2 (neurobasal, 2% B27, 1% penicillin/streptomycin, 1× Glutamax, filter sterilized). Subsequently, cells were filtered through 70 μm Cell Strainer (Corning), counted using Neubauer chamber, plated in 24 well plate (0.13e6 cells/well, DIV 0), and maintained at 37°C. 30% of the media were replaced with fresh media in every 3 days. Experiments were performed at DIV 10. ER stress was induced in primary hippocampal neurons using Tunicamycin (Sigma Aldrich). 2 μg/ml of Tunicamycin was added to primary neuronal culture and incubated for 6 h and compared to those treated with DMSO for same time. To model hypoxia, primary hippocampal neuronal cultures were incubated in normoxia (20% $O_2$) in a standard cell culture incubator for 10 days before they were used in an experiment. For hypoxic conditions, cells were incubated at 1% $O_2$ for 4 h using the *in vivo₂* 400 hypoxia workstation (Baker Ruskin). Cells from the same isolation were kept in normoxia at 20% $O_2$ as a control.

## Quantitative RT–PCR

qPCR (qPCR) primers were designed using Universal probe library Assay Design Center and were purchased from Sigma. Transcriptor High Fidelity cDNA Synthesis Kit (Roche) was used to prepare cDNA. UPL probes were used for quantification, and data were normalized to HPRT1 expression as internal control. Relative gene expression was analyzed by $2^{-\Delta\Delta Ct}$ method. Primers for quantitative PCR analysis:

| Gene | Forward sequence | Reverse sequence |
|---|---|---|
| Bcap31 | tattgctggcttttccttgc | gctgggagatgagagtcacc |
| Fez1 | tgaaaatgactctggcatcaa | gcatcatttcttcaatttcctca |
| Fez2 | cctctcggagaaagggatg | gaatgcatgtccaactgctc |
| Kmt2a | ccacctaaggcaatcagcat | gggggctgtcgatgtttag |
| Kmt2b | caagaacgtccatgctgct | gacaggcagcagccaact |
| Kmt2c | tgaagaggaaaagcaggctatg | gggggtgttggaggattact |
| Kmt2d | atcaggtgaacggacaggtg | gaggctcctgcttgatgc |
| CamkIIδ transgene | ttgaagggtgccatcttgaca | ggtcatgcatgcctggaatc |

## Western blot

Western blot was performed according to previous study (Bahari-Javan et al, 2012). To quantify H3K4me3 and H3K9ac levels, H3K4me3 (Abcam, ab8580; 1:500) antibody and H3K9ac (Abcam, ab4441; 1:1,000) antibody were used, respectively. Unmodified H3 level measured with H3 antibody (Abcam, ab1791, 1:500) was used as internal control.

## Bioinformatics analysis

Bulk RNA-sequencing data analysis has been performed according to Benito et al (2015). Quality of sequencing files for RNA-seq were determined using FastQC. Raw sequencing files were mapped to mouse genome (mm10) using STAR, and read counts were enumerated using featureCounts. Variance stabilizing normalized data were adjusted for effects from biological covariates (gender) and technical covariates (sequencing lane). Surrogate variables were determined by sva, and final linear mixed-effects model was employed using limma in R platform. To determine brain-specific changes in transcriptome after oral administration of Vorinostat, we considered additional biological (body weight, heart weight, ventricle weight, lung weight, tibia length) and technical (birth batch, RNA quality, RNA preparation batch) covariates. Genes with FDR < 0.05 were considered as differentially expressed. For small RNA quantification, sequencing reads were mapped to the mm10 (mouse genome) using mapper.pl function of miRDeep2. Reads shorter than 18 nucleotides were discarded, and known microRNAs were determined using miRDeep2 module. All the operations were performed in default settings. miRBase 22 (http://www.mirbase.org/) database was used to retrieve mature microRNA sequence. A wrapper of the steps applied during mapping and counting is available as package (https://github.com/mdrezaulislam/MicroRNA). Samples with high mapping quality and library size were considered in downstream analysis. Differentially expressed genes or microRNAs were determined using mixed linear model accounting for technical and biological covariates. microRNAs with FDR < 0.05 were considered as differentially expressed.

Biological processes were analyzed using Gene Ontology (http://geneontology.org/). For pathway analysis, KEGG (https://www.genome.jp/kegg/) and Reactome (https://reactome.org/) databases were used. Hypergeometric test analysis was performed using GeneOverlap (http://bioconductor.org/packages/GeneOverlap/). For ChIP-Seq data analysis, quality of the raw data was determined by FastQC. Subsequently, raw data were mapped to the mouse genome (mm10) using Bowtie2. After alignments, files were sorted and mapped reads with MAPQ < 5 were filtered out. Next, alignments that mapped to different location of the genome were removed using samtools. Quality of files after alignment was further examined by deepTools and ChIPQC. Genome-wide narrow peaks were determined using MACS2 (https://github.com/taoliu/MACS). Regions that have been previously associated with high signals regardless of experimental condition were removed, and input normalized peaks were determined using DiffBind. Differential peaks were determined using linear mixed effect model in limma. Peaks with FDR < 0.05 were used for further analysis. Peaks were annotated using ChIPSeeker. ChIP peaks were mapped at promoter (TSS ± 2kb) of genes using ngs.plot. Weighted co-expression analysis for both microRNAs and mRNAs was performed using WGCNA (Langfelder & Horvath, 2008). More specifically, for RNA-seq data analysis, a soft power of 21 was used for correlation analysis, based on fit to scale-free topology. Whole network connectivity was determined using robust "bicor" correlation method. Module size of 150 with ds 2 was used for the analysis. Eigenvalues for each module were determined using moduleEigengenes function. Network of top 30 hub genes was visualized in VISANT (Hu et al, 2005). microRNA co-expression analysis was also performed according to method mentioned above. However, based on fit scale-free topology soft power of 9 was chosen and subsequently module size of 10 with ds 3 was used to carry out module identification and eigenvalue determination. For the identification of genes that host microRNA promoters, microRNA promoters for mouse were retrieved from Fantom 5 database (De Rie et al, 2017). Since these data are originally from old mouse genome (mm9), the genomic co-ordinates were converted to a recent genome (mm10) using UCSC LiftOver. microRNA annotations were converted to updated miRbase 22 using miRBaseConverter (Xu et al, 2018). Ensembl IDs of genes that harbor microRNA promoters were retrieved using BiomaRt. H3K4me3 peaks were mapped to promoter (TSS ± 2kb) of these genes using ngs.plot (Shen et al, 2014). External gene expression datasets that have been used from other studies were downloaded from NCBI GEO (https://www.ncbi.nlm.nih.gov/geo/) and mapped in house to have consistency in results.

## Statistical analysis

Transgenic mice were randomly allocated into the vehicle and Vorinostat groups, and the experimenter was blinded in all experiments. Similarly, wild-type mice were randomly allocated into MI and sham-operated groups and the experimenter was blinded in all subsequent experiments. All the statistical analyses as mentioned in the main text are performed in Prism (version 7.0) or in R.

**The paper explained**

**Problem**

Cognitive dysfunction is a severe burden in heart failure patients. Moreover, these patients have an increased risk for age-associated dementia. The underlying molecular mechanisms are presently unknown, and thus, corresponding therapeutic strategies to improve cognition in heart failure patients are missing.

**Results**

Using mice as model organisms, we show that heart failure leads to memory impairment and a profound alteration of gene expression in hippocampus, a brain region that is important for cognition and is affected early in Alzheimer's disease. Further analysis reveals that heart failure-related down-regulation of hippocampal genes is linked to reduced neuronal H3K4 methylation. These pathological changes are ameliorated via the systemic administration of an epigenetic drug via multiple mechanisms.

**Impact**

International organizations such as the European Society of Cardiology (ESC) recommend that cardiology and dementia research experts should team-up to identify therapeutic interventional options for managing cognitive impairment in subjects with heart failure. Our study provides first insight to the molecular processes by which heart failure contributes to neuronal dysfunction and points to novel therapeutic avenues to treat cognitive defects in heart failure patients.

## Data availability

Raw data for all next-generation sequencing samples and corresponding code can be accesses via the following accession number via Gene Expression Omnibus GSE159449[¶]: https://www.ncbi.nlm.nih.gov/geo/query/acc.cgi?acc = GSE159449

**Expanded View** for this article is available online.

## Acknowledgements

The authors thank Susanne Burkhard, Alessya Kretzschmar, and Sabrina Koszewa for technical support. This work was supported by the funds from the SFB1002 (D04) and TO 822(1-1) of the German Research Foundation (DFG) to KT, and DMK was supported by funds from the IRTG1816 and DFG Ka1269/13-1 of the DFG. AF was supported by the DZNE, the DFG under Germany's Excellence Strategy—EXC 2067/1 390729940, the Hans and Ilse Breuer Foundation, the SPP1738 project FI981/6-2 & FI981/15-1 and the ERC consolidator grant DEPI-CODE (648898). Open Access funding enabled and organized by Projekt DEAL.

## Author contributions

MRI coordinated the project, performed experiment, analyzed data, and wrote the manuscript. DL breed mice and performed echo. MSS performed ChIP-seq. RMH contributed to some behavioral experiments and echo analysis. TB performed tissue dissection. MJM performed qPCR analysis. JC performed immunoblot analysis. MG performed qPCR experiments. EV contributed to the analysis of Barnes Maze data. CS, DMK, and AZ performed hypoxia experiments. FS, KT, and AF designed the project. KT and AF wrote the manuscript and co-supervised all work related to this manuscript.

## Conflict of interest

The authors declare that they have no conflict of interest.

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
