## [Review Process File · EMBO Molecular Medicine]

Epigenetic gene-expression links heart failure to memory impairment

André Fischer, Rezaul Islam, Dawid Lbik, Mohammad Sakib, Raul Hofmann, Tea Berulava, Julia Cha, Vakhtang Ellerdashvili, Christian Schiffmann, Anke Zieseniss, Dörthe Katschinski, Farahnaz Sananbenesi, and Karl Toischer

DOI: [10.15252/emmm.201911900](https://doi.org/10.15252/emmm.201911900)

Corresponding author(s): André Fischer (A.Fischer@eni-g.de) , Karl Toischer (ktoischer@med.uni-goettingen.de)

Review Timeline:

Submission Date:	13th Dec 19
Editorial Decision:	17th Dec 19
Revision Received:	18th Dec 19
Editorial Decision:	19th Dec 19
Revision Received:	15th Jan 20
Editorial Decision:	17th Feb 20
Revision Received:	24th Oct 20
Editorial Decision:	1st Dec 20
Revision Received:	4th Dec 20
Accepted:	11th Dec 20

Editor: Jingyi Hou & Celine Carret

Transaction Report:

(Note: With the exception of the correction of typographic and spelling errors that could be a source of ambiguity, letters and reports are not edited. Depending on transfer agreements, referee reports obtained elsewhere may or may not be included in this compilation. Referee reports are anonymous unless the Referee chooses to sign their reports.)

Thank you for submitting your manuscript "Epigenetic gene-expression links heart failure to memory impairment" to EMBO Molecular Medicine. I have now carefully read your manuscript and discussed it with my colleagues. I regret to say that we find the manuscript not well suited for publication in EMBO Molecular Medicine and therefore have decided not to proceed with peer review.

We appreciate that your data investigate mechanistic links between heart failure and cognitive dysfunction in mice through the means of RNAseq and microRNAseq. The gene expression datasets obtained from the transgenic heart failure mouse model overlap with hypoxia, ER stress and dementia. Behavioural analyses confirm that the Camk1dc transgenic mice present with impaired memory consolidation, and the ChIPseq data point towards hypomethylation in the hippocampus of these mice, while a HDAC inhibitor restored memory and gene expression. We find the topic to be of interest. Unfortunately, because HDAC inhibitors were reported before to be beneficial both in cognitive dysfunction and heart failure conditions, we find that the study doesn't offer the sort of novel translational potentials we like to see. As for the mechanistic insights, they remain correlative at this stage and how exactly heart failure leads to cognitive impairment is still not clear. Therefore, I am afraid that we do not find the study suitable for EMBO Molecular Medicine at this stage.

I notice that you have chosen not to share your work with other EMBO Press editors. However, I very much believe that your study would be a good match for our sister journal Life Science Alliance (LSA). I would then warmly suggest you to transfer your manuscript to them. I would like to emphasise that transferring your work to LSA does not require any reformatting. Please feel free to contact either me or my colleague Andrea Leibfried at LSA for any questions you may have about the transfer.

any thanks for looking at our manuscript. While I understand your decision, I like to briefly comment.

You name two reasons for not sending our manuscript out for review.

You state that Aβ inhibitors were reported to be beneficial for cognitive function and heart failure conditions and conclude that therefore our data would not be novel. Furthermore you state that the ...mechanistic insights, ... remain correlational at this stage and how exactly heart failure leads to cognitive impairment is still not clear.

I believe some confusion might stem from the fact that we are providing interdisciplinary data in a novel heart failure research field.

Indeed, Aβ inhibitors have been proven effective in neurodegenerative diseases. The data on Aβ inhibitors in cardiac diseases is comparable sparse but there is evidence that Aβ inhibitors also ameliorate cardiac pathology. However, these findings are only indirectly linked to our study since they represent pure neurological and cardiological research questions.

We are addressing the issue of cognitive defects in patients with heart failure and aim to elucidate at the molecular level why such patients using model systems have an increased dementia risk. As outlined in our manuscript, I am not aware of any comparable study that has addressed this issue in such way. Thus, there is not a single study so far that demonstrated cognitive effects of oral Aβ inhibitor treatment in a model for heart failure. In any case, these data only provide the insight that aberrant gene-expression plays a key role in cognitive defects seen in response to heart failure and offer a therapeutic strategy. I like to reiterate that in our experimental setting Aβ inhibitor treatment did not affect cardiac hypertrophy.

Mechanistically, we show that the sole manipulation of the heart leads to specific changes in hippocampal gene-expression. We find that these changes reflect the induction of cellular stress pathways including hypoxia followed by the down-regulation of genes important for memory function and are thus most likely direct consequences of hypoxia. We then model hypoxia and H₂O₂-stress in neuronal cultures and observe that induction of these stress pathways are sufficient to drive the down-regulation of learning and memory genes.

Further mechanistic insight could now be provided by manipulating for example hypoxia signaling, e.g. via modulating IGF1 pathways. However, for a therapeutic insight of view such data will not add to our understanding, since the time window for intervention directed to hypoxia will be very limited. Rather, our data suggest that these initial cellular changes manifest at the level of the neuronal epigenome. This finding raises the hope for treatment of patients that already suffer from heart failure, which is the reality in clinical practice.

In other words, we show that a sick heart leads to memory impairment and we provide first mechanistic insight. In turn, we suggest a therapeutic strategy to improve memory function even if the heart has been sick for some time/or still is.

As neuroscientists these research questions had been also new to us and we had many discussions with our colleagues from cardiology but as also mentioned in the cover letter, these are very pressing questions and major scientific innovations specifically ask for this novelty of interdisciplinary research.

This is because, patients suffering from cardiac and cognitive issues will become a major burden in our aging societies. Thus, we feel that our data should be published in a prominent journal.

I understand for your comments that maybe we did not communicate the novelty and importance of our findings in the most optimal manner. I wonder, if you consider a resubmission in case we re-write the manuscript accordingly.

Best regards

Andre

Thank you for asking us to reconsider your manuscript for EMBO Molecular Medicine.

I appreciated your arguments and decided to ask for external advice from one of our board member. Our adviser agreed that the topic was of great interest and recommended sending the paper out for review. Before I do that however, I would like to ask you to modify your article to highlight the novelty in a better way as you suggested you could do. Could you please do that and resubmit your revised manuscript early in January?

The Authors have submitted an updated revision without any further correspondence.

Thank you for the submission of your manuscript to EMBO Molecular Medicine. We have now heard back from the three referees whom we asked to evaluate your manuscript.

You will see that the three referees have very similar and overlapping concerns. Overall, they find the study interesting and original. However, more mechanism needs to be provided to link vorinostat to the neurological effects observed (ref. #1 and #2), the main conclusion should be replicated in a different mouse model of heart failure (ref. #2 and #3), the effect of vorinostat on the heart itself needs to be investigated (ref. #1 and #2) and finally, other epigenetic marks should be looked into, not only H3K4me1 (ref. #3).

Given the interest and clinical relevance of the study, we would like to invite a major revision of this work, and would like to encourage you to address the above issues as highlighted. I realised that revising your article along these lines would result in potentially a lot of work and additional time, for this reason, we have agreed to give you 6-months revision time to start with. Please get in touch after 6-month should you need additional time.

***** Reviewer's comments *****

Referee #1 (Comments on Novelty/Model System for Author):

The paper is interesting in that it showed for the first time the possible involvement of epigenetic changes in mediating the memory dysfunction during heart failure. However, the study is not very mechanistic. The study with an HDAC inhibitor is quite weak and the effect of the drug treatment could be mediated through many other mechanisms besides restoration of euchromatin. I also have serious concerns regarding the data quality and the lack of rigorous statistical evaluation regarding the effect of the HDAC inhibitor upon cognitive function and gene expression. This paper requires substantial amounts of work with more experiments if it is considered further.

Referee #1 (Remarks for Author):

The authors show that, in a mouse model of heart failure, hippocampal expression of genes involved in memory function is decreased most likely due to decreases in H3K4me3 and loss of euchromatin. Treatment with Vorinostat, an HDAC inhibitor, improved cognitive function in heart failure mice possibly through restoration of euchromatin and induction of microRNA clusters. Although the connection between heart failure and cognitive dysfunction has been noted previously, perhaps this paper shows the possible involvement of epigenetic changes in hippocampal neurons. It also shows that administration of an HDAC inhibitor partially restores cognitive function in mice by improving euchromatin.

General

1. Although the authors suggest the involvement of the decreased H3K4me3 and the loss of euchromatin as potential mechanisms mediating the decreased memory function in heart failure mice, experimental evidence to support this notion is not strong at the present form. Vorinostat is an HDAC inhibitor and whether the treatment effectively restores H3K4me3 in genes involved in memory function in hippocampal neurons is not demonstrated. The data showed in expanded view 1 is poor and not convincing. In addition, the action of Vorinostat could be mediated through miRNA or acetylation of non-histone proteins or even in cell types other than hippocampal neurons. The authors could have considered interventions directly and selectively restoring H3K4me in hippocampal neurons.
2. Although the authors claim that Vorinostat does not exhibit any effects upon the heart, the data demonstrated in expanded view 2 show substantial effects upon heart and LV weight. In addition, many lines of evidence from the previous studies suggest that HDAC inhibitors significantly affect cardiac hypertrophy and heart failure. It is unclear whether the effect of Vorinostat is solely mediated through its effects upon the brain.

Specific

1. In Figure 1, the authors should show the index of heart failure, including cardiac output and lung congestion. The authors should clarify the sex and ages of the animals used for the analyses and exactly when the analysis was conducted. A clear rationale for studying at the particular time point is also needed
2. In Figure 3B, the authors should clarify how they sorted neuronal nuclei for ChIP-seq analyses.
3. In Figure 4 and 5, the authors should show the statistical analyses regarding the difference between TG-vehicle and TG-Vorinostat. If there is no significance, one cannot say that Vorinostat normalizes cognitive function or RNAs/microRNAs
4. The authors could have clearer rationales as to why they used the HDAC inhibitors to rescue decreases in H3K4me and investigated miRNA to evaluate the effect of Vorinostat.
5. The authors argue that decreases in H3K4me and decreases in cognitive function in mice caused by decreased perfusion in the brain. It is possible that the mice remain compensated and cardiac output may be maintained. Other mechanisms such as neuro-hormonal compensatory mechanism may drive changes in the brain.

Referee #2 (Remarks for Author):

These are interesting observations because the authors demonstrate that mice with heart failure show certain changes in gene expression in the hippocampus and that an HDAC inhibitor can attenuate an impaired memory function. However, there are several major points that need to be addressed:

1. The authors use a transgenic mouse model with excessive CaMKII overexpression. there are several other heart failure models that recapitulate the pathophysiological causes of heart failure more closely. Do the authors see similar changes in such mouse models (TAC or MI, or genetic cardiomyopathies)?
2. HDAC inhibitors were shown to be cardioprotective. How does it affect cardiac function in the CaMKII model? Did the authors characterize the cardiac phenotype (as done in Fig. 1A-C in mice without vorionstat)? Isn't it possible that the neuronal effects are secondarily mediated by a better cardiac function?
3. Did the authors rule out that transgenic expression of CaMKII may be leaky?
4. Heart failure is typically associated with depression. Did the authors check for depressive behavior (e.g. non helplessness) in CaMKII transgenic mice?
5. What are the signals that mediate heart failure-induced changes in neuronal gene expression. More mechanistic insights with proof of concept experiments would be needed? The authors suggest cerebral hypoperfusion. Can the authors use a model of hypoperfusion, which is independent of heart failure (such as carotid artery occlusion)?

Referee #3 (Comments on Novelty/Model System for Author):

The main issue of this manuscript, from my stand point, relates to the use of only one model of heart failure. Some of the data should be confirmed using another model, in order to exclude that what shown here is the consequence of the unicity of the model used rather than a consequence of HF. I have also raised a question related to the effect of heart failure on epigenetic marks other

than H3K4me1, which could be relevant. I have also to report that I am not a neurobiology expert, rather on epigenetics and heart failure.

Referee #3 (Remarks for Author):

This is an interesting and provocative manuscript, which relates to the effects on behavior and memory of heart failure in terms of epigenetically-mediated gene expression changes in hippocampal cells, linking gene expression modification and behavioral changes. The authors have used a mouse model of cardiac-specific overexpression of CAMK2delta of heart failure. By RNA sequencing the hippocampus, they found a profound modification of gene expression. They also performed ChIP with H3K4 monomethylation, a histone modification mark which identifies active enhancers, therefore linking this histone modification mark with gene expression modifications. They have also used an epigenetic drug, vorinostat, a histone deacetylases inhibitor clinically applied in humans, showing that this drug modifies hippocampal gene expression and ameliorates behavioral changes accordingly.

The manuscript is interesting because it touches upon a clinically very relevant problem, that is, the consequences of heart failure on behavior. To my knowledge, this is the first attempt. There are some significant caveats, which I report below.

1. Only one model of HF, and not the most frequently used one, was employed. Inducing heart failure by overexpression of a gene, in addition a kinase, carries some risks. Extracardiomyocyte leaking expression of the transgene is not infrequent, using this promoter and even if the increase is very limited, an augmented level of a kinase might lead to unwanted effects. Thus, while I do not ask to repeat the whole experimental scheme with another HF model, the authors should show that some molecular events are reproducible in other models, for instance transverse aortic constriction.

2. The authors have assessed the epigenetic consequences of HF on hippocampal cells only though H3K4me1, which is an important epigenetic modification, but is not the only one. Consequences on other epigenetic marks? Are they involved?

Referee #1: (Remarks for Author):**Referee #1. General point 1.**

Referee #1 summarizes in his/her 1st general comment several related points that we like to address individually.

He/she says: *“Although the authors suggest the involvement of the decreased H3K4me3 and the loss of euchromatin as potential mechanisms mediating the decreased memory function in heart failure mice, experimental evidence to support this notion is not strong at the present form. Vorinostat is an HDAC inhibitor and whether the treatment effectively restores H3K4me3 in genes involved in memory function in hippocampal neurons is not demonstrated...”*

In the revised version of our manuscript we have addressed this question experimentally. As suggested by this reviewer, we asked if Vorinostat treatment would affect hippocampal H3K4me3 and therefore performed H3K4me3 ChIP-seq from hippocampal neuronal nuclei of CamkII δ TG that were either treated with vehicle or Vorinostat. Vehicle-treated wild type mice served as an additional control. We were able to reproduce our initial findings and observed that H3K4me3 is decreased in hippocampal neurons of vehicle-treated CamkII δ TG when compared to vehicle-treated wild type control littermates. We show that Vorinostat-treatment increases H3K4me3 in CamkII δ TG mice and thereby provide a feasible explanation for the restored expression of the “learning and memory genes” that were decreased in hippocampal CamkII δ TG mice. These data are now shown within modified panel A and novel panel E of Figure 5 and the corresponding legend. Furthermore, our novel findings are described in the text of the revised manuscript on page 9, lines 14-23. And page 10, lines 26-28.

Referee 1 continues *“...The data showed in expanded view 1 is poor and not convincing....”*

We understand that this question is related to the previous comment on the effect of Vorinostat on H3K4me3. As described above, we now present ChIP-seq data to show that H3K4me3 is increased in response to Vorinostat-treatment and added a more thorough discussion of the data. See page 9, lines 14-23. Please note that former Expanded view figure 1 now became expanded view figure 2.

Referee 1 continues: *“...In addition, the action of Vorinostat could be mediated through miRNA or acetylation of non-histone proteins or even in cell types other than hippocampal neurons...”*

We agree with this comment. This is why we had performed for example smallRNA-sequencing and presented these data in former Figure 5G and H, that now became Fig 5H and I and Expanded View Figure 7. In fact, our data suggest that Vorinostat induces a microRNA response that helps to explain the reinstated expression of RNA module 2 which was increased in CamkII δ TG mice. To elucidate all possible mechanisms by which Vorinostat helps to reinstate – at least in part – gene-expression and memory function in cognitively impaired CamkII δ TG mice is beyond the scope of this manuscript. However, we now address this issue more specifically in our revised manuscript. Please see page 9, last line, page 10, lines 1-8 & 27-28 and page 13, lines 27-34.

Referee 1 continues: *“...The authors could have considered interventions directly and selectively restoring H3K4me in hippocampal neurons. ...”*

We are thankful for this comment. To specifically target H3K4me3 might indeed be a valuable strategy and we are actively working on related projects. For example, a suitable approach would be to use pharmacological compounds that would either activate H3K4-Methyltransferases or inhibit the corresponding De-methylases. However, such compounds are not well characterized. Moreover, the knowledge about the six different H3K4-methyltransferases and the - at least - 4 different de-methylases in the adult brain is only beginning to emerge and we (amongst others) will further contribute to this field (e.g. see Kerimoglu et al., 2017; PMID: 28723559). Any strategy targeted towards the H3K4-enzymes specifically is therefore far away from clinical application. Considering the existing knowledge about HDAC inhibitors as potential drugs to treat cognitive diseases, the

fact that Vorinostat affects neuronal gene-expression and improves memory function when orally delivered, is moreover an FDA-approved drug that is currently tested for treatment of patients with mild-cognitive impairment and can affect H3K4me3 was the basis to choose Vorinostat. In response to this referee's comment we now discuss this issue in greater detail. Please see page 7, lines 21-32 and page 14, lines 7-10 of the revised manuscript.

Referee #1, General point 2

He/she says: *"Although the authors claim that Vorinostat does not exhibit any effects upon the heart, the data demonstrated in expanded view 2 show substantial effects upon heart and LV weight. In addition, many lines of evidence from the previous studies suggest that HDAC inhibitors significantly affect cardiac hypertrophy and heart failure. It is unclear whether the effect of Vorinostat is solely mediated through its effects upon the brain."*

This is an important remark. To address this question, we had presented within former expanded figure 2 data to show that in our experimental setting Vorinostat treatment did not affect body weight, neither did it ameliorate reduced heart or left ventricle weight. We believe that the arrangement of these data might have been misleading, since we presented it in a opposite order as shown in the main Fig. 4. We apologize and corrected this mistake and moreover include the results obtained via pairwise statistical analysis comparing CamKII δ TG Vorinostat and CamKII δ TG Vehicle groups. Moreover, we analyzed additional cardiac parameters and now also demonstrate that Vorinostat had no effect on the ejection fraction, cardiac output and cardiac index. Please note that all of these data are now presented as novel expanded figure 3 and its legend and are described in the text of the revised manuscript on page 8, line 22. We also discuss these data in greater detail and refer to previous findings suggesting that HDAC inhibitors can ameliorate the phenotypes in models for cardiac diseases and point to the fact that in such studies either different models/drugs, higher concentrations or prolonged duration of treatment was used when compared to our study. Please see page on page 13, lines 13-19.

Referee #1 Specific point 1.

He/she says: *"In Figure 1, the authors should show the index of heart failure, including cardiac output and lung congestion. The authors should clarify the sex and ages of the animals used for the analyses and exactly when the analysis was conducted. A clear rationale for studying at the particular time point is also needed."*

We have now added the requested additional measures to Figure 1, panel C. See also page 3 lines 17-18. We also describe now in greater detail the animals used in our study and the rationale for the investigated time points. Please refer to page 3, line 13-16, page 14, lines 22-31.

Referee #1 Specific point 2.

"In Figure 3B, the authors should clarify how they sorted neuronal nuclei for ChIP-seq analyses"

The corresponding description was part of a "supplementary methods" file. We now moved all methods to the main text and describe this procedure in greater detail on page 16, lines 22-35 and page 17, lines 1-14.

Referee #1 Specific point 3.

He/She remarks *"In Figure 4 and 5, the authors should show the statistical analyses regarding the difference between TG-vehicle and TG-Vorinostat. If there is no significance, one cannot say that Vorinostat normalizes cognitive function or RNAs/microRNAs."*

We now added the statistical results from pairwise comparisons (including TG-Vorinostat vs TG-vehicle) to the relevant figure panels. Please refer to Fig 4, Fig 5, novel figure expanded view 3 and the corresponding figure legends.

Referee #1 Specific point 4.

“The authors could have clearer rationales as to why they used the HDAC inhibitors to rescue decreases in H3K4me and investigated miRNA to evaluate the effect of Vorinostat. “

We appreciate this comment that is related to the general comment (4th remark) of this referee. We now take the opportunity to explain our rationale for employing the HDAC Inhibitor Vorinostat in greater detail in the revised version of our manuscript. Please see page 7, lines 22-32 of the revised manuscript. We also discuss in greater detail, why we studied the microRNAs as potential mechanisms. Please see page 10, lines 1-8, page 13, lines 27-34 of the revised manuscript.

Referee #1 Specific point 5.

He/She says: *“The authors argue that decreases in H3K4me and decreases in cognitive function in mice caused by decreased perfusion in the brain. It is possible that the mice remain compensated and cardiac output may be maintained. Other mechanisms such as neuro-hormonal compensatory mechanism may drive changes in the brain.”*

We appreciate this comment and apologize that our statements were misleading. We had discussed decreased perfusion as one possible mechanism but of course there are additional and complimentary explanations. We address this point now in greater detail in the discussion of our manuscript. Please see page 11 lines 7-12 as well as page 4, lines 14-15. We also adapted Fig.6 and its legend.

Referee #2 (Remarks for Author):

Referee #2, Point 1.

He/she say: *“The authors use a transgenic mouse model with excessive CaMKII overexpression. there are several other heart failure models that recapitulate the pathophysiological causes of heart failure more closely. Do the authors see similar changes in such mouse models (TAC or MI, or genetic cardiomyopathies)? “*

In response to this question we have now analyzed an additional model for heart failure. The CamkII δ c TG mice already represent a genetic over-expression model and the TAC model – at least in our view - is less suitable in the context of brain function, since TAC induces elevated blood pressure in the right carotid arteries, while blood pressure would be lower in the left one. Therefore, we followed this referee’s suggestion and employed an MI model. The corresponding data is present within novel expanded view figure 1 and 5. In brief, we find that mice subjected to MI develop the expected cardiac phenotypes. Importantly, MI mice also exhibit hippocampus-dependent memory impairment and hippocampal gene-expression changes that were similar to the ones we observe in CamkII δ c TG mice. We describe and discuss these novel findings in the revised version of our manuscript. Please see novel expanded figures 1 and 5, and its legend, page 6, lines 10-13, page 9, lines 12-14 & lines 25-26 and page 11, lines 7-8.

Referee #2, Point 2.

Referee 2 states *“HDAC inhibitors were shown to be cardioprotective. How does it affect cardiac function in the CaMKII model? Did the authors characterize the cardiac phenotype (as done in Fig. 1A-C in mice without vorinostat)? Isn't it possible that the neuronal effects are secondarily mediated by a better cardiac function? “*

We appreciate this comment that is essentially identical to a remark made by referee 1. Therefore, please refer to our answer to “Referee #1, General point 2” and see novel expanded figure 3 and its legend, page 8, line 22 and page 13, lines 13-19.

Referee #2, Point 3.

“Did the authors rule out that transgenic expression of CaMKII may be leaky?”

Yes. We had presented these data in Fig 1A of the manuscript. We agree, that this observation was not well described in the text and might have been overlooked. Therefore, we now refer more specifically to this point. Please see page 3, lines 9-10 of the revised manuscript.

Referee #2, Point 4.

“Heart failure is typically associated with depression. Did the authors check for depressive behavior (e.g. non helplessness) in CaMKII transgenic mice? “

We did not test for depressive-like behavior. In response to this reviewer’s question we now performed such experiments in 3 months old CamkII δ TG and MI mice (1 month after surgery), hence at the same time points that were analyzed for learning behavior and epigenetic gene-expression. We employed the Porsolt forced swim (PFS) test but failed to detect significant differences amongst groups. While depressive like behavior has not been tested in CamkII δ TG mice, depressive-phenotypes have been observed after MI. However, some studies report an acute effect (e.g. see Wann et al, 2009; PMID 18562428; Frey et al, 2014; PMID 25400562), while other data for example suggest that phenotypes such as learned helplessness are affected at later time points in a subgroup of mice that underwent MI (Bruns et al., 2019; PMID 31025825). In the context our study, we can conclude that depressive-like phenotypes measured via the PFS test in CamkII δ TG and MI mice are not affected at the time point of memory impairment (See Figure below). However, a better understanding of the mechanisms that may underlie the comorbidity of heart failure, depression and cognitive impairment is highly interesting but beyond the scope of our study.

For now, we decided to simply describe our findings in the discussion of the revised manuscript, since our manuscript already includes 5 main and 7 expanded view figures. Please see page 11 lines 30-34 and page 12, lines 1-5. We are however happy to generate a corresponding expanded view figure upon the editor’s or referee’s request.

Figure. Porsolt forced swim (PFS) test in 3 months old CamkII δ TG and MI. Left panel: 3 month old CamkII δ TG and control littermates were subjected to the PFS test, as described in one of our previous studies (Agis-Balboa, 2017; PMID 28768717). The time spent mobile within the 5 min duration of the test was not significantly different amongst groups (unpaired *t*-Test; *P* = 0,39, *n* = 7/group). **Right panel:** 3 months old MI (*n*=8) and corresponding sham control mice (*n*=10; one month after surgery) were subjected to PFS. The time spent mobile within the 5 min duration of the test was not significantly different amongst groups (unpaired *t*-Test; *P* = 0,1). Please note the mobility time in the corresponding control groups, hence the WT vs. sham

groups differed. We like to stress the fact that this is not uncommon, since these animals were obtained from different breeding colonies and moreover all mice in the MI experiment underwent surgery while this was not the case in the experimental group consisting of CamkII δ TG and control mice. It is therefore important to stress, that a meaningful comparison is only possible within each experiment. ns, non-significant

Referee #2, Point 5.

He/she says: “What are the signals that mediate heart failure-induced changes in neuronal gene expression. More mechanistic insights with proof of concept experiments would be needed? The authors suggest cerebral hypoperfusion. Can the authors use a model of hypoperfusion, which is independent of heart failure (such as carotid artery occlusion)? “

This remark is in part related to the comment made by Referee #1, Specific point 5. Please refer to our response to this point. In brief, we mention now that hypoperfusion is only one possible mechanism and discuss now also alternative explanations. Please see page 11 lines 7-12 as well as page 4, lines 14-15. We also adapted Fig.6 and its legend accordingly. We appreciate the suggestion to consider models for cerebral hypoperfusion that would not involve heart failure. Although in this study, we were specifically interested to study the effect of heart failure on cognitive function, such type of experiments are certainly part of our future research in which we like to elucidate the impact of hypoperfusion vs. circulating factors but we feel that at present such data would be beyond the scope of our manuscript.

Referee #3 (Remarks for Author):

Referee #3, Point 1.

He/she says "Only one model of HF, and not the most frequently used one, was employed. Inducing heart failure by overexpression of a gene, in addition a kinase, carries some risks. Extracardiomyocyte leaking expression of the transgene is not infrequent, using this promoter and even if the increase is very limited, an augmented level of a kinase might lead to unwanted effects. Thus, while I do not ask to repeat the whole experimental scheme with another HF model, the authors should show that some molecular events are reproducible in other models, for instance transverse aortic constriction. "

We agree with this comment that addresses essentially the same question as raised by Referee #2, Point 1. Please refer to our response to Referee #2, Point 1.

Referee #3, Point 2.

"The authors have assessed the epigenetic consequences of HF on hippocampal cells only though H3K4me1, which is an important epigenetic modification, but is not the only one. Consequences on other epigenetic marks? Are they involved?"

This is a valuable question. We do not exclude that other histone-modifications play a role in the observed gene-expression changes. In fact, it is very likely that other histone-marks are altered in the hippocampus of CamkII δ c TG mice. However, H3K4me3 is a key histone-mark linked to eu-chromatin formation and active gene-expression and is thus a suitable general marker to study epigenetic gene-expression control. Moreover, we decided to analyze H3K4me3, since "methylation" was observed as a key pathway represented by the genes down-regulated in the hippocampus of CamkII δ c TG mice. In line with this, we found that these down-regulated genes were significantly enriched for genes that are decreased in the hippocampus of mice lacking the important H3K4 Methyltransferase Kmt2a. We apologize that we did not address these points in greater detail in the previous version of the manuscript. We do this now. Please refer to page 12, lines 20-23 & 32, page 13, lines 27-34

Thank you for the submission of your revised manuscript to EMBO Molecular Medicine. We have now received the enclosed report from the three referees who were asked to re-assess it. As you will see the referees are overall supportive and I am pleased to inform you that we will be able to accept your manuscript pending the following amendment s: 1. All three referees still raised a series of - mostly minor - concerns on your work, which need to be addressed and/or discussed.

***** Reviewer's comment s *****

Referee #1 (Comment s on Novelty/Model System for Author):

The revised manuscript is improved significantly. The paper report s novel and important information. I have some suggestion s for the author s.

Referee #1 (Remark s for Author):

The author s are quite responsive to my comment s, and the paper is improved significantly. Many of the data now support s the author s' statement in this work.

The author s should normalize heart weight and Left ventricular weight with either body weight or tibial length.

Principle component analysis should be principal component analysis.

The author s elected not to modulate H3K4me3 directly due to technical reason s. As the author s stated, Vorinostat induces increased histone acetylation that generally promotes euchromatin formation. Thus, one cannot exclude the possibility that Vorinostat improves cognitive dysfunction through H3K4me3-independent action s. This can be stated as a limitation in this study.

Referee #2 (Remark s for Author):

the author s did additional experimental work to address my previous concern s. The manuscript has

improved. However, I still have one comment. Vorinostat did not affect ejection fraction but it seems to me that hypertrophy is reduced (although it does not reach significance). Since remodeling is associated with secretion of factors, I still think this should be considered and discussed in 1-2 sentences. Secretome analysis from diseased cardiomyocytes would be beyond the current scope but a worthwhile mechanism to look at in the future.

Referee #3 (Comments on Novelty/Model System for Author):

The topic is innovative, that is the link between brain and heart in disease. Heart failure is frequently accompanied by changes of behavior and by depression. The mechanisms in terms of gene expression and possible epigenetic changes have not been addressed so far. This work is one of the first that I know of to try and answer this question. Although there are some limitations (mostly on mechanisms and model used), which other reviewers and I have underlined in the first revision, the manuscript is provocative and under many points, innovative. Limitations should be clearly stated in the discussion though.

Referee #3 (Remarks for Author):

The authors have reviewed the original manuscript by adding some new experiments. In particular, they have added a new model of HF, myocardial infarction, in order to confirm the data obtained with the cardiac-specific CAMK over expression. The data included to answer this point are not as extensive as the one of the transgenic model, but suggest that modifications of the activity of epigenetic regulators might take place also in other instances. Incidentally, I do not understand why the TAC model is not suitable. In terms of mechanisms, the authors refer to hypoperfusion as a possible trigger of epigenetic modifications in the hippocampal area. This seems speculative at this time, while I am not sure why the authors have not taken into account modifications of the sympathetic nervous system or other mechanisms. Limitations of the study in my view should be clearly stated in the discussion, including points raised by other and this reviewer in the two revisions.

Referee #1 (Remarks for Author):

He/she says: *"The authors are quite responsive to my comments, and the paper is improved significantly. Many of the data now supports the authors' statement in this work. The authors should normalize heart weight and Left ventricular weight with either body weight or tibial length."*

We have addressed this issue now within revised Fig 1C and its legend and changed the text accordingly. Please see page 3, lines 17-18.

"Principle component analysis should be principal component analysis."

We corrected this typo. Please see page 3 and 21.

"The authors elected not to modulate H3K4me3 directly due to technical reasons. As the authors stated, Vorinostat induces increased histone acetylation that generally promotes euchromatin formation. Thus, one cannot exclude the possibility that Vorinostat improves cognitive dysfunction through H3K4me3-independent actions. This can be stated as a limitation in this study. "

We agree and had addressed this issue in our previously revised manuscript. Please see page 13, line 34 and page 14, lines 1-5.

Referee #2 (Remarks for Author):

He/she states *"the authors did additional experimental work to address my previous concerns. The manuscript has improved. However, I still have one comment. Vorinostat did not affect ejection fraction but it seems to me that hypertrophy is reduced (although it does not reach significance). Since remodeling is associated with secretion of factors, I still think this should be considered and discussed in 1-2 sentences. Secretome analysis from diseased cardiomyocytes would be beyond tzeurrent scope but a worthwhile mechanism to look at in the future."*

This is an excellent comment and we discuss this issue now in the revised manuscript. Please see page 13, lines 19-21.

Referee #3 (Remarks for Author):

He/she says: *"The authors have reviewed the original manuscript by adding some new experiments. In particular, they have added a new model of HF, myocardial infarction, in order to confirm the data obtained with the cardiac-specific CAMK over expression. The data included to answer this point are not as extensive as the one of the transgenic model, but suggest that modifications of the activity of epigenetic regulators might take place also in other instances. Incidentally, I do not understand why the TAC model is not suitable. In terms of mechanisms, the authors refer to hypoperfusion as a possible trigger of epigenetic modifications in the hyppocampal area. This seems speculative at this time, while I am not sure why the authors have not taken into account modifications of the sympathetic nervous system or other mechanisms. Limitations of the study in my view should be clearly stated in the discussion, including points raised by other and this reviewer in the two revisions."*

We appreciate this comment and understand that the first part of the question is related to the fact that we did not consider the TAC but rather the MI model as a non-genetic approach to study heart failure-induced cognitive dysfunction. We had outlined in the previous revision why we did not consider TAC as our first choice (please see our previous response to referee 2, point 1) but we agree

that it would be worth to test this option experimentally. Thus, we now specifically refer to TAC in the discussion of the revised manuscript. Please see page 11, lines 8-9.

Regarding the idea that hypo-perfusion may not be the sole mechanisms for the observed brain-specific alterations, we completely agree with this reviewer and had already addressed this issue in our previous revision. Please see our response to referee #1, specific point 4 of the previous revision letter.

Accepted

11th Dec 2020

We are pleased to inform you that your manuscript is accepted for publication and is now being sent to our publisher to be included in the next available issue of EMBO Molecular Medicine.

YOU MUST COMPLETE ALL CELLS WITH A PINK BACKGROUND ↓
PLEASE NOTE THAT THIS CHECKLIST WILL BE PUBLISHED ALONGSIDE YOUR PAPER

Corresponding Author Name: Andre Fischer , Karl Toischer

Manuscript Number: EMM-2019-11900